# Estimating Heterogeneous Treatment Effects by Combining Weak Instruments and Observational Data

**Miruna Oprescu**
Cornell University
`amo78@cornell.edu`

**Nathan Kallus**
Cornell University
`kallus@cornell.edu`

## Abstract

Accurately predicting conditional average treatment effects (CATEs) is crucial in personalized medicine and digital platform analytics. Since the treatments of interest often cannot be directly randomized, observational data is leveraged to learn CATEs, but this approach can incur significant bias from unobserved confounding. One strategy to overcome these limitations is to leverage instrumental variables (IVs) as latent quasi-experiments, such as randomized intent-to-treat assignments or randomized product recommendations. This approach, on the other hand, can suffer from low compliance, *i.e.*, IV weakness. Some subgroups may even exhibit zero compliance, meaning we cannot instrument for their CATEs at all. In this paper we develop a novel approach to combine IV and observational data to enable reliable CATE estimation in the presence of unobserved confounding in the observational data and low compliance in the IV data, including no compliance for some subgroups. We propose a two-stage framework that first learns *biased* CATEs from the observational data, and then applies a compliance-weighted correction using IV data, effectively leveraging IV strength variability across covariates. We characterize the convergence rates of our method and validate its effectiveness through a simulation study. Additionally, we demonstrate its utility with real data by analyzing the heterogeneous effects of 401(k) plan participation on wealth.

## 1 Introduction

The use of observational data for individual-level causal analyses is becoming increasingly common in personalized medicine, online platforms, and any setting where understanding individualized responses is crucial and/or presents an opportunity for personalization. The key quantity for such analyses is the conditional average treatment effect (CATE), which captures how treatment effects vary according to baseline covariates (features). This measure provides insight into effect heterogeneity and enables personalization.

Using observational data can nonetheless introduce bias from unobserved confounding, where the observed relationship between outcomes and interventions is influenced not only by treatment effects but also by variables that influence both outcome and treatment, such as socioeconomic status, health, user mood, *etc.*, which are not captured by baseline covariates. These biases can skew causal effect estimates, resulting in unreliable analyses or even harmful policy decisions.

Randomized trials are the gold standard for causal inference, but they are often infeasible. For instance, digital services cannot force users to view or buy a product, and clinical trials cannot require invasive treatments. A common alternative is to randomize the *encouragement* of certain actions, such as recommending a product or treatment. These encouragements can serve as instrumental variables (IVs) which, under certain conditions, enable unbiased estimation of treatment effects [4].

Identification of CATEs using IVs crucially hinges on the premise that compliance – the correlation between the treatment received and the intent/encouragement – is nonzero across all baseline-covariate

values. When compliance is nonzero but small, IV-based estimates tend to exhibit high variance, making them unreliable [3]. In practice, the assumption of strong compliance is often violated. For example, users on digital platforms may ignore recommendations entirely or reject certain types of content, while participants on mobile health platforms may disregard prompts (*e.g.* taking 250 steps per hour) due to time constraints or lack of interest.

To address the challenge of estimating unbiased CATEs in the presence of unobserved confounding and low IV compliance, we introduce a two-stage framework. In the first stage, we estimate a biased, confounded CATE from observational data. Then, in the second stage, we utilize an IV to learn the confounding bias by weighting the samples according to their compliance levels. By assuming only that the bias can be extrapolated, this approach extends treatment effect adjustments even to groups minimally influenced by the IV, employing a transfer learning approach that leverages varying instrument strengths across covariate groups.

This framework mirrors strategies in causal inference that combine randomized trials with observational data to address low covariate overlap. Building on this body of work, we introduce two methodologies for extrapolating confounding bias within the observational dataset: a parametric estimation approach, assuming the confounding bias adheres to a parametric form, and a transfer learning strategy that assumes a shared representation between the true and biased CATE. We study the properties of our CATE estimators in finite samples and validate our approaches through comprehensive empirical studies.

## 2 Related Work

We briefly overview related work here; for a more comprehensive discussion, refer to Appendix A.

**Heterogeneous treatment effect estimation from observational data:** Recent advances in machine learning have expanded the use of observational data to estimate CATEs using diverse techniques such as random forests [51], Bayesian algorithms [24], deep learning [48], and meta-learners [33]. However, these methods often unrealistically assume an absence of confounding, limiting their real-world applicability. Efforts to account for unobserved confounding either construct *bounds* on treatment effects [17, 40] or use latent variable models and multiple/sequential treatments to debias CATE estimates [9, 36, 53], but they frequently depend on unverifiable assumptions or require accurate proxy data, reducing their practical utility.

**Heterogeneous treatment effect estimation using IVs:** Integrating machine learning with instrumental variable (IV) methods enhances CATE estimation flexibility over traditional approaches. Techniques range from advanced two-stage least squares (2SLS) that incorporate complex feature mappings via kernel methods [49] and deep learning [54] to neural networks for conditional density estimation [21] and moment conditions for IV estimation [8]. Yet, these rely on the consistent relevance of instruments across covariate groups, which is not guaranteed with weak instruments.

**Treatment Effect Estimation with Weak Instruments:** Traditional IV methods like 2SLS can be unreliable when instruments are weak, leading to biased, high-variance estimates. Recent advancements include novel estimators such as bias-adjusted 2SLS, limited information maximum likelihood, and jackknife IV estimators (see [25] and references therein). Other techniques attempt to reduce variance by exploiting first-stage heterogeneity (variation in compliance) [1, 13]. Some approaches also combine multiple weak instruments into robust composites, useful in settings like genetic studies [30]. Our approach extends [1, 13] by leveraging compliance weighting to estimate heterogeneous effects and address weak instruments using additional observational data.

**Combining observational and randomized data:** Increasing research focuses on integrating observational datasets with randomized control trial (RCT) data to mitigate observational bias. Strategies include imposing structural assumptions, such as strong parametric constraints [29], or assuming a shared structure between biased and unbiased CATE functions [23], as well as optimizing dual estimators from both data types for improved bias correction [55]. Our work aligns with efforts to debias treatment effects using both observational and experimental data, but also addresses challenges such as low IV compliance, the need to debias the overall effect function rather than individual outcome functions, and the complexity of estimating CATEs from IV data using a ratio estimator.

**Where our work lies:** To the best of our knowledge, no current estimation technique effectively combines an IV study, particularly one with weak instruments or low compliance, with an observational

study to derive robust and unbiased CATE estimates. We bridge this gap by introducing two robust and consistent CATE estimation techniques, building upon previous work on combining RCT and observational data [23, 29], as well as work that addresses the complexities associated with weak instruments [1, 13].

## 3 Background and Setup

We consider the standard setting of causal inference where the goal is to estimate the conditional average treatment effect of a binary treatment $A \in \{0, 1\}$ on an outcome $Y \in \mathbb{R}$ in the presence of covariates $X \in \mathcal{X} \subseteq \mathbb{R}^m$. Our approach is grounded in Rubin's potential outcomes framework, wherein each unit is associated with two potential outcomes $Y(0), Y(1)$ of which only $Y = Y(A)$ is observed (causal consistency). Our objective is to learn the CATE function, which is given by:

$$\tau(x) = \mathbb{E}[Y(1) - Y(0) \mid X = x]. \tag{1}$$

However, we only have access to $n_O$ i.i.d. samples from an observational dataset $O = (X_i^O, A_i^O, Y_i^O)_{i=1}^{n_O} \sim (X^O, A^O, Y^O)$. Thus, we face the fundamental problem of causal inference: only the outcome under the administered treatment is observed, while the counterfactual remains unobserved. Without further assumptions, there exists the possibility of unobserved confounding, leading to a situation where

$$\tau^O(x) = \mathbb{E}[Y^O \mid A^O = 1, X^O = x] - \mathbb{E}[Y^O \mid A^O = 0, X^O = x] \neq \tau(x), \tag{2}$$

which indicates a persistent bias in the observed treatment effects that does not diminish even with an increasing sample size. We denote this bias by $b(x)$, that is:

$$b(x) = \tau(x) - \tau^O(x).$$

Assuming this bias is induced by a set of unobserved confounders $U \subseteq \mathbb{R}^k$, the discrepancy arises because the selection into treatment in the observational population is influenced by $U$, which also impacts the outcome $Y^O$. Our goal is to mitigate this bias by leveraging additional data.

Alongside the observational dataset, we have $n_E$ i.i.d. samples from an experimental, intent-to-treat dataset $E = (X_i^E, Z_i^E, A_i^E, Y_i^E)_{i=1}^{n_E} \sim (X^E, Z^E, A^E)$ where $Z^E$ is a binary instrument taking values in $\{0, 1\}$. We let $X^E \in \mathcal{X}$ and assume the $p_{X^E}(x) = p_{X^O}(x)$, where $p_X$ denotes the density of the random variable $X$. Moreover, we assume that the joint distribution of covariates and unobserved confounders $(X, U)$ is consistent across both datasets. As before, we use $Y^E(A, Z)$ to denote the potential outcome given treatment $A$ and instrument $Z$. Additionally, let $A^E(Z)$ denote the potential treatment under instrument $Z$, and define the compliance and defiance indicators $C$ and $D$ by $C := \mathbb{I}[A^E(1) > A^E(0)]$ and $D := \mathbb{I}[A^E(1) < A^E(0)]$, respectively. We assume that this dataset follows standard IV assumptions on the data generating process:

**Assumption 1** (Standard IV Assumptions). *We assume the following properties hold: (Exclusion)* $Y^E(A, Z) = Y^E(A)$, *i.e. the instrument affects the outcome only through the treatment; (Independence)* $Z \perp\!\!\!\perp U \mid X$ *for any unobserved confounder $U$; and (Relevance) there exists a subset $\mathcal{X}' \subseteq \mathcal{X}$ with non-zero measure such that $Z^E \not\perp\!\!\!\perp A^E \mid X^E$ for $X^E \in \mathcal{X}'$.*

**Assumption 2** (Unconfounded Compliance [52]). *The individual treatment effect is independent of the compliance status given covariates:* $Y^E(1) - Y^E(0) \perp\!\!\!\perp (A^E(1) - A^E(0)) \mid X^E$.

We note that the relevance assumption in Assumption 1 is a weaker version of the standard IV assumptions since we allow for arbitrarily weak instruments in some regions of the covariate spaces. With Assumption 1 and Assumption 2, we can identify the CATE for $x \in \mathcal{X}'$ as:

$$\tau^E(x) = \frac{\mathbb{E}[Y^E \mid Z^E = 1, X^E = x] - \mathbb{E}[Y^E \mid Z^E = 0, X^E = x]}{\mathbb{E}[A^E \mid Z^E = 1, X^E = x] - \mathbb{E}[A^E \mid Z^E = 0, X^E = x]} := \frac{\delta_Y(x)}{\gamma(x)} = \tau(x). \tag{3}$$

We provide the proof of Equation 3 in Appendix B. Here, $\gamma(x)$ denotes heterogeneous compliance, a measure of instrument strength, given by $\gamma(x) = P(C = 1 \mid X^E = x) - P(D = 1 \mid X^E = x)$ under Assumption 2. A *strong* instrument ($\gamma(x) \to 1$) indicates high adherence to the recommended treatment, with $\gamma(x) = 1$ signifying perfect compliance, similar to a true randomized controlled trial. Conversely, a *weak* instrument ($\gamma(x) \to 0$) suggests minimal influence on treatment uptake, with $\gamma(x) = 0$ indicating no compliance and a confounded selection into treatment. The relevance

assumption in Assumption 1 ensures $\gamma(x) \neq 0$ for $x' \in \mathcal{X}'$, validating the estimation procedure in Equation 3. However, small $\gamma(x)$ values lead to estimates of $\tau(x)$ with high asymptotic variance. Moreover, we wish to extend the $\tau(x)$ estimation from $\mathcal{X}'$ to $\mathcal{X}$, our population of interest.

Thus, relying solely on observational data results in biased $\tau(x)$ estimates, while experimental data alone can yield high variance or invalid estimates for $x \in \mathcal{X}$ with low compliance. This work addresses these challenges by strategically combining the strengths of both datasets to provide a robust CATE estimation technique.

**Notation:** We denote the $L_2$ norm of a function $f$ as $\|f\|_{L_2} := \mathbb{E}_F[f(X)^2]^{1/2}$, and the $L_2$ Euclidean norm of a vector $\theta \in \mathbb{R}^d$ as $\|\theta\|_2$. The notation $\widehat{f}$ represents the estimated value of a parameter or function, where $f$ is the true value. We omit the distribution subscript when clear from context; *e.g.*, $\mathbb{E}[X^E]$ and $\mathbb{E}[X^O]$ denote expectations over experimental and observational samples, respectively.

# 4 Estimation Method

To obtain robust estimates of the CATE function for the population of interest $\mathcal{X}$, we propose a two-step framework that integrates information from both the observational data and the IV study. First, we estimate the confounded CATE function $\widehat{\tau}^O(x)$ using the observational data $(X_i^O, A_i^O, Y_i^O)_{i=1}^{n_O}$. This is a well-established problem in both causal inference and machine learning, and it can be addressed using various existing techniques, including meta-learners ([33]), random forests ([51]), and neural networks ([48]).

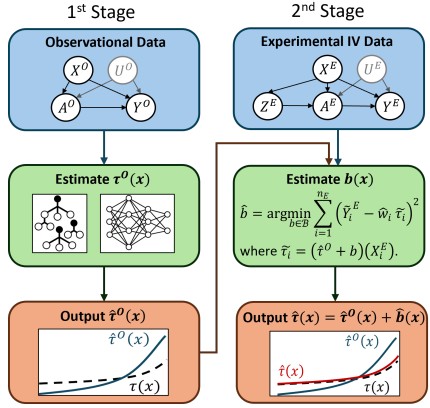

Figure 1: Illustration of our two-stage procedure: the first stage learns a biased CATE from observational data, while the second stage uses IV data to correct the bias.

Next, we wish to approximate the bias function $b(x) = \tau(x) - \tau^O(x)$ using the learned $\widehat{\tau}^O(x)$. Without oracle access to the true CATE function $\tau(x)$, we instead rely on samples from the experimental (IV) study $(X_i^E, Z_i^E, A_i^E, Y_i^E)_{i=1}^{n_E}$ for which we can estimate an unbiased, though potentially high variance, CATE for $x \in \mathcal{X}'$, as given in Equation 3. Our approach hinges on the following lemma:

**Lemma 1.** *[CATE Estimation with IVs] Let* $\pi_Z(x) := P(Z^E = 1 \mid X^E = x)$ *be the instrument propensity. Then, the following identity holds for every* $x \in \mathcal{X}'$:

$$\mathbb{E}\left[ \frac{Y^E Z^E}{\pi_Z(x)\gamma(x)} - \frac{Y^E(1 - Z^E)}{(1 - \pi_Z(x))\gamma(x)} \middle| X^E = x \right] = \tau(x)$$

We note that in the case of randomized instrument assignment, the instrument propensity is known and often given by a constant, *i.e.*, $\pi_Z(x) = \pi_Z > 0$. By defining $V_Z(x) := \pi_Z(x)(1 - \pi_Z(x))$, Lemma 1 shows that the bias function $b(x)$ can be expressed in terms of observable quantities as $b(x) = \mathbb{E}\left[ \frac{Y^E Z^E(1 - \pi_Z(X^E)) - Y^E(1 - Z^E)\pi_Z(X^E)}{V_Z(X^E)\gamma(X^E)} - \tau^O(x) \mid X^E = x \right]$ for $x \in \mathcal{X}'$. This formulation suggests that we can estimate $\gamma(x)$ and, if necessary, $\pi_Z(x)$ from data and utilize the pseudo-outcome

$$\frac{\widetilde{Y}^E}{\widehat{V}_Z(X^E)\widehat{\gamma}(X^E)} := \frac{Y^E Z^E(1 - \widehat{\pi}_Z(X^E)) - Y^E(1 - Z^E)\widehat{\pi}_Z(X^E)}{\widehat{V}_Z(X^E)\widehat{\gamma}(X^E)}$$

along with the estimated $\widehat{\tau}^O(x)$, in a subsequent regression task to obtain an unbiased and consistent estimate of $b(x)$ for $x \in \mathcal{X}'$ (provided $\pi_Z$, $\gamma$, and $\widehat{\tau}^O$ are estimated consistently). However, such an estimator only provides estimates for $\mathcal{X}'$ where $\gamma(x) \neq 0$. Additionally, for small values of $\gamma(x)$, $\pi_Z(x)$, and $1 - \pi_Z(x)$, this method may result in high variance in the estimates $\widehat{b}(x)$, especially for certain parametric function classes. To address these challenges, we weight the data samples by the inverse variance of $\widetilde{Y}^E/(\widehat{\gamma}(x)\widehat{V}_Z(x))$ given by $\text{Var}(\widetilde{Y}^E|X^E = x)^{-1}\widehat{\gamma}^2(x)\widehat{V}_Z^2(x)$. This approach is frequently used in generalized least squares methods (GLS, [2]) to confer the algorithm asymptotic efficiency. While $\text{Var}(\widetilde{Y}^E|X^E = x)$ can be estimated from data using machine learning methods, it is generally preferable to weight the estimator solely by compliance and instrument propensity to

---
**Algorithm 1** CATE Estimation with Parametric Extrapolation
---
1: **Input:** Observational dataset $O = (X_i^O, A_i^O, Y_i^O)_{i=1}^{n_O}$, IV dataset $E = (X_i^E, Z_i^E, A_i^E, Y_i^E)_{i=1}^{n_E}$, $\tau^O(x)$
 estimator $\mathcal{T}$, $\gamma(x)$ estimator $\mathcal{G}$, $\pi_Z(x)$ estimator $\mathcal{P}$, known mapping $\phi : \mathcal{X} \to \mathbb{R}^d$.
2: Learn $\widehat{\tau}^O(x)$ using $\mathcal{T}$ on $O$. Let $\widetilde{\mathbf{Y}} \in \mathbb{R}^{n_E}, \widetilde{\mathbf{X}} \in \mathbb{R}^{n_E \times d}$.
3: **for** $k = 1, 2, \ldots, K$ **do**
4:      Set $\mathcal{I}_k = \{i \in \{1, \ldots, n_E\} : i = k - 1 \pmod K\}$.
5:      Use data $\{(X_i^E, Z_i^E, A_i^E, Y_i^E) \in E : i \notin \mathcal{I}_k\}$ to learn $\widehat{\gamma}^{(k)}(x)$ with $\mathcal{G}$ and $\widehat{\pi}_Z^{(k)}(x)$ with $\mathcal{P}$.
6:      **for** $i = k - 1 \pmod K$ **do**
7:          Set $\widetilde{\mathbf{Y}}_i = Y_i^E Z_i^E (1 - \widehat{\pi}_Z^{(k)}(X_i^E)) - Y_i^E (1 - Z_i^E) \widehat{\pi}_Z^{(k)}(X_i^E) - \widehat{w}^{(k)}(X_i^E) \widehat{\tau}^O(X_i^E)$.
8:          Set $\widetilde{\mathbf{X}}_{ij} = \widehat{w}^{(k)}(X_i^E) \phi_j(X_i^E)$ for each $j \in \{1, \ldots, d\}$.
9: **Output:** $\widehat{\theta} = (\widetilde{\mathbf{X}}^T \widetilde{\mathbf{X}})^{-1} \widetilde{\mathbf{X}}^T \widetilde{\mathbf{Y}}$.
---

avoid issues with small values of $\mathrm{Var}(\widetilde{Y}^E | X^E = x)$. Assuming the bias function belongs to a class of functions $\mathcal{B}$, our proposed algorithm can be described by the following weighted empirical risk minimization (ERM) procedure.

$$\widehat{b} = \arg\min_{b \in \mathcal{B}} \sum_{i=1}^{n_E} \left( \widetilde{Y}_i^E - \widehat{\gamma}(X_i^E) \widehat{V}_Z(X_i^E) \widehat{\tau}^O(X_i^E) - \widehat{\gamma}(X_i^E) \widehat{V}_Z(X_i^E) b(X_i^E) \right)^2 \tag{4}$$

where the factor $\widehat{\gamma}^2(x) \widehat{V}_Z^2(x)$ was used for weighting the squared loss. This estimator automatically extrapolates to all of $\mathcal{X}$ since we assign weights of 0 when $\widehat{\gamma}(x) = 0$. Moreover, this method places higher emphasis on lower-variance pseudo-outcomes, thereby minimizing the risk of overfitting to data points with high variance. This weighting technique is commonly employed in other IV estimation tasks, such as local *average* treatment effect estimation (LATE), where weighting data points by compliance yields estimators with lower variance ([1, 13]).

The weighting scheme in Equation 4 creates a weighted distribution, $\tilde{p}_{X^E}(x)$, for optimizing the ERM procedure. Since $\tilde{p}_{X^E}(x)$ differs from the target distribution $p_{X^E}(x)$, this introduces a transfer learning problem. Without additional constraints on the function class $\mathcal{B}$, the minimization in Equation 4 may yield many possible solutions. To ensure a unique or limited solution set, $\mathcal{B}$ must have low complexity or require further structural assumptions. We explore two function classes $\mathcal{B}$: a parametric class defined by $b(x) = \theta^T \phi(x), \theta \in \mathbb{R}^d$ with a known mapping $\phi : \mathcal{X} \to \mathbb{R}^d$, and a second parametric class where $b(x) = \nu^T \phi(x)$, with $\nu \in \mathbb{R}^d$ and $\phi \in \Phi$ being a learned representation common to both the observational and IV datasets.

### 4.1 Integrating Observational and Experimental Data via Parametric Extrapolation

We consider a parametric class $\mathcal{B}_\phi = \{\theta^T \phi(x) : \theta \in \mathbb{R}^d\}$ for a known mapping $\phi : \mathcal{X} \to \mathbb{R}^d$. Since the compliance factor $\gamma(x)$, instrument propensity $\pi_Z(x)$, and the parameter of interest $\theta^T$ are learned from the same dataset $E$, we propose the following $K$-fold cross-fitted estimation procedure:

$$\widehat{\theta} = \arg\min_{\theta \in \mathbb{R}^d} \sum_{k=1}^K \sum_{i \in \mathcal{I}_k^E} \left( \widetilde{Y}_i^E - \widehat{w}^{(k)}(X_i^E) \widehat{\tau}^O(X_i^E) - \theta^T \widehat{w}^{(k)}(X_i^E) \phi(X_i^E) \right)^2 \tag{5}$$

where $\widehat{w}^{(k)}(X_i^E) := \widehat{\gamma}^{(k)}(X_i^E) \widehat{V}_Z^{(k)}(X_i^E)$, and the compliance factor $\widehat{\gamma}^{(k)}$ and instrument propensity $\widehat{\pi}_Z^{(k)}, k \in [K]$ are trained on $E$ excluding the $k^{\text{th}}$ fold containing indices $\mathcal{I}_k^E$. K-fold cross-fitting is crucial because it ensures that the weights are learned from data distinct from that used in the ERM algorithm. This separation is essential for maintaining desirable theoretical properties as we remain methodologically agnostic to the techniques used for learning $\gamma$ and $\pi_Z$.

The compliance factor $\gamma(x) = \mathbb{E}[A^E \mid Z^E = 1, X^E = x] - \mathbb{E}[A^E \mid Z^E = 0, X^E = x]$ can be estimated using standard machine learning classification algorithms, either by training separate classifiers for $A^E \mid Z^E = 1, X^E = x$ and $A^E \mid Z^E = 0, X^E = x$ or by using one classifier with $Z^E$ as an additional feature. Similarly, instrument propensity estimation is a straightforward classification task with $Z^E$ as the target. Given estimates $\widehat{\tau}^O, \widehat{\gamma}^{(k)}$, and $\widehat{\pi}_Z^{(k)}$, the result in Equation 5 is obtained by performing an OLS procedure with the targets $\widetilde{Y}_i^E - \widehat{w}^{(k)}(X_i^E) \widehat{\tau}^O(X_i^E)$ and the

design matrix $\widetilde{\mathbf{X}} = W(X^E)\Phi(X^E)$. Here, $W(X^E) = \operatorname{diag}(\widehat{w}^{(k)}(X_i^E), \dots, \widehat{w}^{(k)}(X_{n_E}^E))$, and $\Phi(X^E) = (\phi(X_1^E), \dots, \phi(X_{n_E}^E))^T$. The two-step procedure is detailed in Algorithm 1.

Next, we provide theoretical guarantees for our parametric extrapolation approach. We begin by describing the regularity assumptions that enable the consistency of our estimator.

**Assumption 3** (Regularity Assumptions). *The following claims are true:*

1. *(Treatment Positivity in O)* $\epsilon \leq P(A^O = 1 \mid X^O = x) \leq 1 - \epsilon$ *for some* $\epsilon > 0$.

2. *(Instrument Positivity in E)* $\epsilon \leq \pi_Z(X^E), \widehat{\pi}_Z(X^E) \leq 1 - \epsilon$ *for some* $\epsilon > 0$.

3. *(Boundedness)* $Y^E$, $Y^O$, $\|X^E\|_2$, $\|\phi(X^E)\|_2$, $\widehat{\tau}^O(x)$, $\widehat{\gamma}(x)$ *are uniformly bounded.*

4. *(Realizability of $b(x)$)* $b(x) \in \mathcal{B}_\phi$, *i.e.* $\tau(x) - \tau^O(x) = \theta^T\phi(x)$ *for some* $\theta \in \mathbb{R}^d$.

5. *(Identifiability of $\theta$)* $\mathbb{E}[\phi(X^E)\phi(X^E)^T]$ *is invertible.*

The first two conditions in Assumption 3 are standard in causal inference, ensuring that both treatments (or instruments) and controls are observable for every $x \in \mathcal{X}$, enabling CATE estimation. The third condition imposes a common boundedness assumption to control the growth of estimands. The fourth condition ensures our model for the bias function $b(x)$ is well-specified given $\mathcal{B}_\phi$. The final condition requires that the design matrix has rank $d$, ensuring we can learn the parameter $\theta$ from data. Given Assumption 3, we present the following theoretical result:

**Theorem 2** (Estimator Consistency for Parametric Extrapolation). *Let $r_\gamma(n)$, $r_{\pi_Z}(n)$, and $r_{\tau^O}(n)$ be $o_p(1)$ functions of $n \in \mathbb{N}$ such that $\|\gamma - \widehat{\gamma}^{(k)}\|_{L_2} \leq r_\gamma(n_E)$, $\|\pi_Z - \widehat{\pi}_Z^{(k)}\|_{L_2} \leq r_{\pi_Z}(n_E)$, and $\|\tau^O - \widehat{\tau}^O\|_{L_2} \leq r_{\tau^O}(n_O)$. Furthermore, assume the conditions of Assumption 1, Assumption 2, and Assumption 3 hold. Then, the parameter $\widehat{\theta}$ returned by Algorithm 1 is consistent and satisfies*

$$\left\|\widehat{\theta} - \theta\right\|_2 = O_p\left(r_\gamma(n_E) + r_{\pi_Z}(n_E) + r_{\tau^O}(n_O) + 1/\sqrt{n_E}\right).$$

*Moreover, $\widehat{\tau}$ is consistent on $\mathcal{X}$ with convergence rate given by*

$$\|\widehat{\tau} - \tau\|_{L_2} = O_p\left(r_\gamma(n_E) + r_{\pi_Z}(n_E) + r_{\tau^O}(n_O) + 1/\sqrt{n_E}\right).$$

We include the proof of Theorem 2 in Appendix B. The core insight is that weighted OLS remains consistent as long as the estimates for $\widehat{\gamma}$, $\widehat{\pi}_Z$, and $\widehat{\tau}^O$ are themselves consistent. However, the overall convergence rate is constrained by the slowest of these rates. In most cases, $\pi_Z$ is assumed to be known, meaning the convergence rate is primarily dictated by the rates of $\widehat{\gamma}$ and $\widehat{\tau}^O$. This result highlights the trade-off involved in leveraging both datasets to achieve accurate effect estimation for the target population.

**Remark 1** (Impact of Realizability Violations). *When realizability does not hold, i.e. $b(x) \notin \mathcal{B}$, our estimator may be inconsistent and exhibit asymptotic bias, proportional to the deviation of the true function from $\mathcal{B}$. Nonetheless, conducting this analysis might still be valuable, as the resulting bias may be smaller than confounding bias in observational estimates or the variance from low compliance in IV studies. Thus, even with uncertain realizability, our method may provide more accurate CATE estimates by effectively balancing bias and variance.*

### 4.2 Integrating Observational and Experimental Data via a Common Representation

Without expert knowledge, the mapping $\phi(x)$ may not be known a priori. In this section, we introduce a method to jointly learn both the unbiased CATE function and the mapping $\phi(x)$ (hereafter referred to as the *representation*), based on the assumption that the true CATE $\tau(x)$ and the biased CATE $\tau^O(x)$ share a common representation. This approach leverages machine learning techniques that assume a common structure across tasks, such as multi-task and transfer learning. In causal inference, it has been suggested that a shared representation can be assumed between treatment arms [47, 48] or between randomized data and confounded observational data [23]. This framework enables us to learn the bias function $b(x)$ even when the mapping $\phi(x) \in \Phi$ is otherwise unknown.

We consider a class $\Phi$ of representations $\phi(x) : \mathcal{X} \to \mathbb{R}^d$ and assume that there exists a shared representation $\phi \in \Phi$ between the true and biased CATEs. Specifically, there exist linear hypotheses $h, h^O \in \mathbb{R}^d$ such that $\tau(x) = h^T\phi(x)$ and $\tau^O(x) = (h^O)^T\phi(x)$, resulting in the bias function

---
**Algorithm 2** CATE Estimation with Representation Learning
---
1: **Input:** Observational dataset $O = (X_i^O, A_i^O, Y_i^O)_{i=1}^{n_O}$, IV dataset $E = (X_i^E, Z_i^E, A_i^E, Y_i^E)_{i=1}^{n_E}$, $(\phi, h^O)$ estimator $\mathcal{T}$, $\gamma(x)$ estimator $\mathcal{G}$, $\pi_Z(x)$ estimator $\mathcal{P}$.
2: Learn $\widehat{\phi}(x)$ and $\widehat{h}^O$ using $\mathcal{T}$ on $O$.
3: Call Algorithm 1 with $\phi = \widehat{\phi}$ and $\widehat{\tau}^O(x) = (\widehat{h}^O)^T \widehat{\phi}(x)$. Let $\widehat{\nu}$ be its output.
4: **Output:** $\widehat{\nu}$.
---

$b(x) = (h - h^O)^T \phi(x) := \nu^T \phi(x)$. For simplicity, we focus on linear-in-representation classes, but more complex hypotheses $h$ with $\tau(x) = h(\phi(x))$ can be considered – see [23, 47]. Thus, $b(x) \in \mathcal{B}_\phi$ for the unknown $\phi$, with $\mathcal{B}_\phi$ defined in Section 4.1. Suppose there exists an ERM algorithm $\mathcal{T}$ that can jointly learn $\phi(x)$ and $h^O$ from the observational data, $O$. Our learning algorithm proceeds as follows: first, we use $\mathcal{T}$ to learn $\widehat{\phi}(x)$ and $\widehat{h}^O$ from $O$, alongside estimates $\widehat{\gamma}^{(k)}(x)$ and $\widehat{V}_Z^{(k)}(x)$ from $E$ as described in Section 4.1. In the second stage, we apply the following ERM procedure to estimate the parameter $\nu$:

$$\widehat{\nu} = \arg\min_{\nu \in \mathbb{R}^d} \sum_{k=1}^K \sum_{i \in \mathcal{I}_k^E} \left( \widetilde{Y}_i^E - (\widehat{h}^O)^T \widehat{w}^{(k)}(X_i^E) \widehat{\phi}(X_i^E) - \nu^T \widehat{w}^{(k)}(X_i^E) \widehat{\phi}(X_i^E) \right)^2. \quad (6)$$

This procedure is detailed in Algorithm 2. Finally, we recover $\widehat{\tau}(x)$ by setting $\widehat{\tau}(x) = (\widehat{h}^O + \widehat{\nu})^T \widehat{\phi}(x)$.

**Example 1** (Representation learning with neural networks). *Let $\Phi$ be a class of feed-forward neural networks. Then $\widehat{\phi}(x), \widehat{h}^O$ and $\widehat{\tau}^O(x)$ can be jointly learned by composing $\Phi$ with two linear output heads for $Y^O \mid A^O = 1, X^O = x$ and $Y^O \mid A^O = 0, X^O = x$, respectively. By taking the difference between the two output heads, we can reconstruct $\widehat{\tau}^O(x)$, assuming that $\mathbb{E}[Y^O \mid A^O = 1, X^O = x]$ and $\mathbb{E}[Y^O \mid A^O = 0, X^O = x]$ are also linear in $\phi$ (see [47, 48]). Without this assumption, we can learn $\tau^O(x)$ directly by composing $\Phi$ with one linear output layer and considering the pseudo-outcome $\frac{Y^O A^O}{\pi_A(X^O)} - \frac{Y^O (1 - A^O)}{(1 - \pi_A(X^O))}$. Here, $\pi_A(X^O) = P(A^O = 1 \mid X^O)$ is the treatment propensity in $O$ and can be learned using any black-box machine learning classifier.*

With this setup, we obtain theoretical results similar to those in Theorem 2:

**Theorem 3** (Estimator Consistency for Shared Representation Learning). *Let $r_\gamma(n)$, $r_{\pi_Z}(n)$, and $r_\phi(n)$ be $o_p(1)$ functions of $n \in \mathbb{N}$ such that $\|\gamma - \widehat{\gamma}^{(k)}\|_{L_2} \leq r_\gamma(n_E)$, $\|\pi_Z - \widehat{\pi}_Z^{(k)}\|_{L_2} \leq r_{\pi_Z}(n_E)$, and $\|\phi - \widehat{\phi}\|_{L_2} \leq r_\phi(n_O)$. Additionally, assume $\|\widehat{\phi}\|_2$ is bounded and $\mathbb{E}[\widehat{\phi}(X)\widehat{\phi}(X)^T]$ is invertible. Let us also consider the conditions specified in Assumption 1 and Assumption 2 to be satisfied. Moreover, assume that $\tau^O(x) = (h^O)^T \phi(x)$ for some $\phi$ that is realizable within the representation class $\Phi$ and let Assumption 3 hold for $\phi$. Under these conditions, the parameter $\widehat{\nu}$ returned by Algorithm 2 is consistent and satisfies*

$$\|\widehat{\nu} - \nu\|_2 = O_p\left(r_\gamma(n_E) + r_{\pi_Z}(n_E) + r_\phi(n_O) + 1/\sqrt{n_E} + 1/\sqrt{n_O}\right).$$

*Moreover, $\widehat{\tau}$ is consistent on $\mathcal{X}$ with convergence rate given by*

$$\|\widehat{\tau} - \tau\|_{L_2} = O_p\left(r_\gamma(n_E) + r_{\pi_Z}(n_E) + r_\phi(n_O) + 1/\sqrt{n_E} + 1/\sqrt{n_O}\right).$$

We provide the proof of Theorem 3 in Appendix B. This result hinges on the realizability assumption in $\Phi$ and the linear-in-representation structure of both $\tau$ and $\tau^O$. In Example 1, $r_\phi(n)$ bounds the generalization error for feed-forward neural networks. For ReLU activations and bounded outputs, $r_\phi(n) = C\sqrt{WL \log W \log n}/\sqrt{n}$, where $W$ is the total number of weights, $L$ is the number of layers, and $C$ is a constant independent of $n$ and $W$ [15, 56]. While this rate is parametric, it scales linearly with $W$, which becomes problematic for over-parameterized networks. For 1-Lipschitz activations and bounded weights, [18] derive a rate of $r_\phi(n) = C\sqrt{\Pi_{l=1}^L M(l)}/n^{1/4}$, where $M(l)$ bounds the Frobenius norm of layer $l$'s weight matrix.

**Practical Guidance in High-Dimensional Settings:** When $\phi(x)$ is high-dimensional, controlling the complexity of $\mathcal{B}_\phi$ through regularization is crucial, especially since the bias function $b(x)$ is used to extrapolate the CATE into low-variance regions where compliance is low and the risk of overfitting is high. In the parametric extrapolation approach (Section 4.1), applying $L_1$ or $L_2$ regularization via

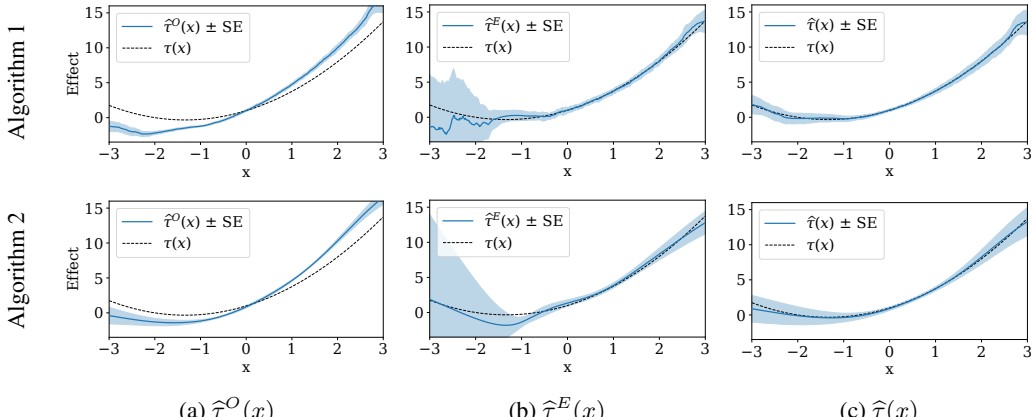

(a) $\widehat{\tau}^{O}(x)$           (b) $\widehat{\tau}^{E}(x)$           (c) $\widehat{\tau}(x)$

Figure 2: Means and standard errors of estimates from 100 simulated dataset pairs $(O, E)$ using Random Forest (top) or Neural Network (bottom) learners. (2a): Biased observational CATE $\tau^{O}(x)$. (2b): High variance CATEs from the IV dataset using Equation 3. (2c): CATEs from Algorithm 2 using parametric extrapolation (top) or representation learning (bottom).

Lasso or Ridge regression in the final step is effective for controlling model complexity. In the shared representation approach (Example 1), regularization not only helps control the parameters $h^{O}$ and $\nu$ but also prevents over-parametrization in the neural network $\phi$. The choice between $L_1$ and $L_2$ regularization, and how they are applied, should be aligned with the data-generating process and the specific characteristics of the model.

## 5 Experimental Results

We apply our method to both simulated and real-world data. First, we use the confounded synthetic data example from [29], along with a similar data generating process (DGP) to simulate an IV study, maintaining the same confounding structure and treatment effects. Using this DGP, we evaluate Algorithm 1 and Algorithm 2 in estimating the unbiased CATE by integrating these datasets. Next, we demonstrate our estimators on a real-world dataset examining the impact of 401(k) participation on financial wealth. Additional experiments, as well as details on model implementation, hyperparameter selection, and validation procedures are in Appendix C. The replication code is available at `https://github.com/CausalML/Weak-Instruments-Obs-Data-CATE`.

### 5.1 Simulation Studies

We generate the observational dataset $O = (X^{O}, A^{O}, Y^{O})$ as follows (see [29])[1]:

$$X \sim \mathcal{N}(0,1), \quad A \sim \text{Bern}(0.5), \quad U \mid X, A \sim \mathcal{N}\left(X\left(A - 0.5\right), 0.75\right)$$
$$Y = 1 + A + X + 2AX + 0.5X^2 + 0.75AX^2 + U + 0.5\epsilon_Y, \quad \epsilon_Y \sim \mathcal{N}(0,1) \tag{7}$$

In this DGP, the true CATE is given by $\tau(x) = 0.75x^2 + 2x + 1$, whereas the biased observational CATE is represented by $\tau^{O}(x) = 0.75x^2 + 3x + 1$. This results in a bias $b(x) = -x$, which is linear in $x$. We modify this DGP to generate the experimental IV dataset $E = (X^{E}, Z^{E}, A^{E}, Y^{E})$ as follows:

$$X \sim \mathcal{N}(0,1), \quad Z \sim \text{Bern}(0.5), \quad A^* \sim \text{Bern}(0.5)$$
$$\gamma(X) = \sigma(2X), \quad C \sim \text{Bern}(\gamma(X)), \quad A = C \cdot Z + (1 - C) \cdot A^*$$
$$U \mid X, A, C \sim C \cdot \mathcal{N}(0,1) + (1 - C) \cdot \mathcal{N}(X\left(A - 0.5\right), 0.75)$$

where $C$ is the (unknown) compliance indicator, $\sigma$ is the logistic sigmoid and we keep the same outcome function as in Equation 7. In this modified DGP, the randomized instrument has compliance sharply determined by $X$, with low $X$ values indicating almost no compliance and high $X$ values indicating near-perfect compliance.

---

[1]For experimental results using a higher-dimensional version of this DGP, refer to Appendix C.

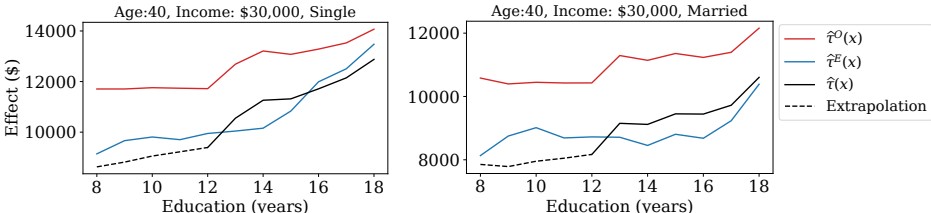

Figure 3: Impact of 401(k) participation on net worth by education level: Using $\widehat{\tau}(x)$ from Algorithm 1, we fix age, income, and binary variables, varying education and marital status. The black line shows results from Algorithm 1, and the dashed line indicates predictions in the no-compliance region. $\widehat{\tau}^O(x)$ is the biased observational CATE, while $\widehat{\tau}^E(x)$ is the IV CATE without non-compliance.

We generate 100 observational and IV datasets, each with 5,000 samples, from the proposed DGP. We first apply Algorithm 1 to each dataset. With a randomized instrument, $\pi_Z(x) = 0.5$. We estimate $\gamma(x)$ as the difference between Random Forest (RF) classifiers trained on $(X^E, A^E) \mid Z^E = 0$ and $(X^E, A^E) \mid Z^E = 1$, *i.e.* one is trained the subset of data where the instrumental variable $Z^E = 0$ (using $X^E$ and $A^E$ as inputs), and another on the subset where $Z^E = 1$. The biased observational CATE is modeled using the T-learner approach [33], with RF regressors trained on $X^O, Y^O \mid A^E = 0$ and $X^O, Y^O \mid A^O = 1$. For comparison, we implement a CATE estimator for the experimental data using Equation 3. We compute $\delta_Y(x)$ as the difference between RF regressors trained on $X^E, Y^E \mid Z^E = 0$ and $X^E, Y^E \mid Z^E = 1$, then divide by $\widehat{\gamma}(x)$, clipping the compliance score at 0.1. We calculate $\widehat{\gamma}(x)$, $\widehat{\tau}^O(x)$, and $\widehat{\tau}^E(x)$ for each dataset pair and proceed with the second step of Algorithm 1 by setting $\phi(x) = x$.

In Figure 2 (top row), we depict the means and standard errors of our estimators across 100 simulations. The first two plots illustrate the learned observational CATE $\widehat{\tau}^O(x)$ and the learned IV CATE $\widehat{\tau}^E(x)$. As expected, $\widehat{\tau}^O(x)$ shows clear bias, while $\widehat{\tau}^E(x)$ has high variance despite aggressive compliance score clipping. The third plot presents the results from Algorithm 1, showing that the resulting $\widehat{\tau}(x)$ is both unbiased and has low variance across $\mathcal{X}$. These findings demonstrate that our two-stage estimation procedure effectively leverages the strengths of both datasets to capture the true CATE and address the limitations of each individual study design.

We note that in our DGP, $\tau(x)$, $\tau^O(x)$, and $b(x)$ are linear in the polynomial representation $(x, x^2)$. Thus, we next apply Algorithm 2 with Example 1 to learn the true CATE and the common representation from the generated dataset. For consistency, we employ feed-forward neural networks (NNs) to estimate all quantities. The estimator for $\widehat{\gamma}$ uses a NN with a sigmoid activation in the output layer, trained on $X^E$ with the pseudo-outcome $2A^E Z^E - 2A^E(1 - Z^E)$. The representation $\phi(x)$ and the biased CATE $\tau^O(x)$ are learned using a representation network with two output heads for learning $Y^O \mid X^O, A^O = 0$ and $Y^O \mid X^O, A^O = 1$. A similar dual-head approach is used to learn $\delta_Y(x)$, by modeling $Y^E \mid X^E, Z^E = 0$ and $Y^E \mid X^E, Z^E = 1$ simultaneously. When calculating $\delta_Y(x)/\gamma(x)$, we clip the compliance score at 0.1. Unlike Algorithm 1, we don't guarantee the polynomial representation will be fully captured by the chosen representation class, but we expect a sufficiently flexible $\Phi$ to adequately represent these relationships.

The means and standard errors of our estimators from 100 simulations using neural networks and Algorithm 2 are shown in Figure 2 (bottom row). As before, $\widehat{\tau}^O(x)$ shows bias, while $\widehat{\tau}^E(x)$ has high variance in low-compliance regions, despite compliance score clipping. However, Figure 2c shows that the $\widehat{\tau}$ returned by Algorithm 2 remains unbiased with relatively low variance across $\mathcal{X}$. This demonstrates that combining observational and IV data, where the biased and true CATE share a representation, allows us to reliably learn both the representation and the unbiased CATE, overcoming the limitations of each individual study.

## 5.2    Impact of 401(k) Participation on Financial Wealth

We demonstrate our method's effectiveness with a real-world case study on the impact of 401(k) participation on financial wealth, using data from the 1991 Survey of Income and Program Participation [11]. The dataset includes 9,915 respondents with nine covariates: age, income, education, family size, marital status, two-earner status, pension status, IRA participation, and home ownership. The primary variable of interest is 401(k) participation ($A$), with eligibility ($Z$) as the instrumental

variable. Although 401(k) eligibility is not randomly assigned, it is argued to maintain conditional independence given observed features [11, 43]. We assume 401(k) eligibility influences net worth only through 401(k) participation, characterizing this as an IV study with one-sided non-compliance, where non-eligible individuals cannot participate ($A^E(0) = 0$). The target variable ($Y$) is net financial assets, calculated as the total of 401(k) balance, bank account balances, and interest-earning assets, minus non-mortgage debt.

To replicate the scenario in this paper, we split the dataset into two halves: one for the IV study and the other for the observational study (where we intentionally remove the instrument information). Our goal is to use these datasets, along with the parametric extension approach in Algorithm 1, to recover the unbiased CATEs. Due to one-sided non-compliance, the estimated compliance factor $\widehat{\gamma}(x)$ is high ($0.49 - 0.90$, see Appendix C). To show the utility of our method, we introduce artificial non-compliance by setting $\gamma(x)$ to 0 for individuals with less than 12 years of education (13% of the population). In the first stage of Algorithm 1, we use RF regressors and classifiers to estimate $\tau^O(x)$, $\gamma(x)$, and $\pi_Z(x)$, with hyperparameters set based on other related work on this dataset [12]. In the second stage, we define the mapping $\phi(x)$ with an intercept term, the 9 covariates, and their interactions (46 features total). We apply a mild $L_1$ regularization in the final linear regression due to the large number of resulting features.

In Figure 3, we study how the CATE function from Algorithm 1 varies with education. We focus on education as it is selected as a top feature by the compliance model, while the outcome models do not rank it as highly significant (see Appendix C). To explore this relationship, we vary education and marital status, holding age and income at their median values and setting all binary variables to zero. Since compliance in the IV study is high, we consider the estimate $\widehat{\tau}^E(x)$ *without* the artificial non-compliance as the ground truth for comparison. Our analysis shows that observational data treatment effects are upwardly biased, likely due to unobserved confounders such as financial literacy. The $\widehat{\tau}(x)$ from Algorithm 1, shown with a dashed line for non-compliance regions, closely aligns with $\widehat{\tau}^E(x)$ (excluding the artificial non-compliance). This demonstrates that combining IV and observational data can effectively estimate unbiased CATEs in real-world settings, offering a robust solution for causal inference even in the presence of low compliance and unobserved confounding.

## 6   Conclusion

This study introduces a method that combines observational and instrumental variable (IV) data to address unobserved confounders in observational studies and low compliance in IV studies. Our two-stage framework first estimates biased CATEs from the observational data, then corrects them using compliance-weighted IV samples. We explore two variations of our procedure: one that models confounding bias parametrically, and another that leverages a shared representation between the true and biased CATEs. Both methods are shown to be consistent, validated through simulations and real-world applications. Our approach holds significant promise for applications in digital platforms, personalized medicine, and economics, offering a robust tool for deriving reliable, actionable insights from complex data. Limitations of our work are discussed in Appendix D.

**Acknowledgements**

We thank the anonymous reviewers for their valuable feedback and insightful suggestions. This material is based upon work supported by the National Science Foundation under Grant No. 1846210 and by the U.S. Department of Energy, Office of Science, Office of Advanced Scientific Computing Research, under Award Number DE-SC0023112.

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

# A   Extended Literature Review

**Heterogeneous treatment effect estimation from observational data:**   Recently, there has been a significant interest in applying machine learning to estimate CATEs using observational data. This field has seen adaptations of a wide range of machine learning techniques, from random forests [39, 51] and Bayesian algorithms *e.g.* [20, 24] to deep learning [5, 27, 48] and blackbox meta-learners [33, 38] that utilize efficient influence functions [14, 31] and Neyman orthogonality [12, 16]. Despite these advancements, a significant challenge remains as these methods typically assume the absence of confounding in observational data (ignorability) – an often unrealistic and unverifiable assumption – limiting their real-world applicability. Without ignorability, point identification of effects is impossible, although some studies propose methods to construct *bounds* on treatment effect estimates under assumptions about the structure of unobserved confounding [17, 28, 40, 45]. Nonetheless, these bounds often have limited practical utility. Other efforts to address confounding bias in CATE estimation rely on latent variable models to recover unobserved confounders from noisy proxies [34, 36] or utilize multiple or sequential treatments [9, 22, 53]. However, these methods also have limited practical impact, as they depend on either the availability of additional accurate proxy data or unverifiable assumptions such as no unobserved single-cause confounders.

**Heterogeneous treatment effect estimation using IVs:**   Machine learning techniques have recently been integrated with instrumental variable methods, offering significant advantages over traditional approaches, including the flexible estimation of CATEs. [49] and [54] expand on two-stage least squares (2SLS) to incorporate complex feature mappings via kernel methods and deep learning. In the same vein, [21] introduced a two-stage neural network for conditional density estimation, while [8] applied moment conditions for IV estimation. [50] propose novel IV estimators that exhibit Neyman orthogonality. However, these techniques rely on the assumption that instruments are relevant across all covariate groups, a condition that is not consistently met with weak instruments.

**Treatment effect estimation with weak instruments:**   Weak instruments compromise the reliability of traditional IV methods like 2SLS, often producing biased, high-variance estimates and undermining causal claims. To mitigate these issues, several approaches have been developed, including bias-adjusted 2SLS estimators, limited information maximum likelihood (LIML), and jackknife instrumental variable (JIVE) estimators (see [25] and references therein). Recent methods reduce 2SLS estimator variance by exploiting first-stage heterogeneity (variation in compliance) through a weighting scheme, as detailed in [1, 13, 32]. However, these methods do not extend to estimating conditional average treatment effects. Another strand of research focuses on combining multiple weak instruments into a robust composite, showing promise in genetic studies using Mendelian randomization ([30, 35]). These approaches require access to multiple weak instruments for the same treatment. Our work aligns most closely with [1, 13, 32] in that we leverage compliance heterogeneity and employ compliance weighting to merge IV studies with observational data for efficient confounding bias estimation. Unlike these studies, however, our approach distinctively estimates heterogeneous effects and leverages additional observational data to address challenges posed by weak instruments.

**Combining observational and randomized data:**   There has been a proliferation of research in combining observational datasets with randomized control trials – experimental data with *perfect* compliance – to mitigate bias from observational studies. One of the strategies is to impose structural assumptions such as strong parametric assumptions for the confounding bias [29] or a shared structure between the biased and unbiased CATE functions that can be estimated from the two datasets [23]. Other studies advocate for dual estimators from both data types, optimizing bias correction through a weighted average [10, 46, 55]. Additionally, approaches like those by [6] and [26] leverage outcomes from different time-steps, such as short-term and long-term effects, to enhance estimation accuracy. Our work is closest to [29] and [23]. However, our study faces additional complexities: firstly, the CATE estimation techniques differ between the datasets, requiring us to debias the overall effect function rather than just individual outcome functions. Secondly, RCTs may not represent the target population due to their narrow scope, our instrumental variable (IV) study faces representation issues due to minimal or absent compliance in strata that are not known a priori. Thirdly, the CATE estimation in our IV study uses a ratio estimator, which is highly sensitive to changes in the compliance denominator, adding a layer of complexity to our analysis.

# B    Proofs of Theorems and Lemmas

## B.1    Proof of Equation 3

The exclusion and independence conditions in Assumption 1 imply that the following identification equation holds:

$$\mathbb{E}\left[Y^E\left(A^E(1)\right) - Y^E\left(A^E(0)\right) \mid X^E = x\right] \tag{8}$$
$$= \mathbb{E}[Y^E \mid Z^E = 1, X^E = x] - \mathbb{E}[Y^E \mid Z^E = 0, X^E = x].$$

By noting that

$$\begin{cases} Y^E\left(A^E(1)\right) - Y^E\left(A^E(0)\right) = Y(1) - Y(0), & \text{when } A^E(1) = 1, A^E(0) = 0 \text{ (compliers)} \\ Y^E\left(A^E(1)\right) - Y^E\left(A^E(0)\right) = Y(0) - Y(1), & \text{when } A^E(1) = 0, A^E(0) = 1 \text{ (defiers)} \\ Y^E\left(A^E(1)\right) - Y^E\left(A^E(0)\right) = 0, & \text{when } A^E(1) = 1, A^E(0) = 1 \text{ (always-takers)} \\ Y^E\left(A^E(1)\right) - Y^E\left(A^E(0)\right) = 0, & \text{when } A^E(1) = 0, A^E(0) = 0 \text{ (never-takers)} \end{cases}$$

the left-hand side of this equation can further be written as:

$$\mathbb{E}[Y^E(A^E(1)) - Y^E(A^E(0)) \mid X^E = x] \tag{9}$$
$$= \mathbb{E}[(Y^E(1) - Y^E(0))(A^E(1) - A^E(0)) \mid X^E = x]$$
$$= \mathbb{E}[Y^E(1) - Y^E(0) \mid X^E = x] \cdot \mathbb{E}[A^E(1) - A^E(0) \mid X^E = x] \quad \text{(Assumption 2)}$$
$$= \tau(x) \cdot (\mathbb{E}[A^E \mid Z^E = 1, X^E = x] - \mathbb{E}[A^E \mid Z^E = 0, X^E = x]) \quad \text{(Assumption 1)}$$

Since the claim of Equation 3 holds for $x \in \mathcal{X}'$, we have that $\mathbb{E}[A^E \mid Z^E = 1, X^E = x] - \mathbb{E}[A^E \mid Z^E = 0, X^E = x] \neq 0$ by the relevance condition in Assumption 1. From Eqs. 8 and 9, we obtain:

$$\tau(x) = \frac{\mathbb{E}[Y^E \mid Z^E = 1, X^E = x] - \mathbb{E}[Y^E \mid Z^E = 0, X^E = x]}{\mathbb{E}[A^E \mid Z^E = 1, X^E = x] - \mathbb{E}[A^E \mid Z^E = 0, X^E = x]}$$

for $x \in \mathcal{X}'$.

## B.2    Proof of Lemma 1

Recall that for any $x \in \mathcal{X}'$, we have that $\gamma(x) \neq 0$ by Assumption 2. Then, assuming that $\pi_Z(x) > 0$, we use the law of total expectation as follows:

$$\mathbb{E}\left[\frac{Y^E Z^E}{\pi_Z(x)\gamma(x)} - \frac{Y^E(1 - Z^E)}{(1 - \pi_Z(x))\gamma(x)}\middle| X^E = x\right]$$
$$= \mathbb{E}\left[\frac{Y^E Z^E}{\pi_Z(x)\gamma(x)} - \frac{Y^E(1 - Z^E)}{(1 - \pi_Z(x))\gamma(x)}\middle| Z^E = 1, X^E = x\right] P(Z^E = 1 \mid X^E = x)$$
$$+ \mathbb{E}\left[\frac{Y^E Z^E}{\pi_Z(x)\gamma(x)} - \frac{Y^E(1 - Z^E)}{(1 - \pi_Z(x))\gamma(x)}\middle| Z^E = 0, X^E = x\right] P(Z^E = 0 \mid X^E = x)$$
$$= \mathbb{E}\left[\frac{Y^E}{\pi_Z(x)\gamma(x)}\middle| Z^E = 1, X^E = x\right] \pi_Z(x)$$
$$- \mathbb{E}\left[\frac{Y^E}{(1 - \pi_Z(x))\gamma(x)}\middle| Z^E = 0, X^E = x\right] (1 - \pi_Z(x))$$
$$= \frac{\mathbb{E}\left[Y^E \mid Z^E = 1, X^E = x\right] - \mathbb{E}\left[Y^E \mid Z^E = 0, X^E = x\right]}{\gamma(x)}$$
$$= \frac{\mathbb{E}\left[Y^E \mid Z^E = 1, X^E = x\right] - \mathbb{E}\left[Y^E \mid Z^E = 0, X^E = x\right]}{\mathbb{E}\left[A^E \mid Z^E = 1, X^E = x\right] - \mathbb{E}\left[A^E \mid Z^E = 0, X^E = x\right]} = \tau(x) \quad \text{(Equation 3)}$$

where the intermediate steps follow from the definitions of $\pi_Z(x)$ and $\gamma(x)$ and the last equality comes from the identification result in Equation 3.

## B.3 Proof of Theorem 2

For simplicity, we omit the $E$ subscripts from $X^E, Z^E, A^E, Y^E$ throughout this proof. Furthermore, assume that $n_E$ is an integer multiple of the number of folds $K$. Let $\widehat{\mathbb{E}}_k f(Z) = \frac{1}{|\mathcal{I}_k|} \sum_{i \in \mathcal{I}_k} f(Z_i)$, recalling that $\mathcal{I}_k = \{i \in \{1, \ldots, n_E\} : i = k - 1 \pmod{K}\}$, which indexes the subset of data in the $k^{\text{th}}$ fold. Then, we can write the estimated parameter $\widehat{\theta}$ as:

$$\widehat{\theta} = \left( \frac{1}{K} \sum_{k=1}^{K} \widehat{\mathbb{E}}_k \left[ \widehat{w}^{(k)}(X)^2 \phi(X) \phi(X)^T \right] \right)^{-1}$$

$$\cdot \frac{1}{K} \sum_{k=1}^{K} \widehat{\mathbb{E}}_k \left[ \left( YZ(1 - \widehat{\pi}_Z^{(k)}(X)) - Y(1 - Z)\widehat{\pi}_Z^{(k)}(X) - \widehat{w}^{(k)}(X)\widehat{\tau}^O(X) \right) \widehat{w}^{(k)}(X)\phi(X) \right]$$

We also define the following quantities:

$$\widetilde{\theta}_{n_E} = \widehat{\mathbb{E}}_{n_E} \left[ w(X)^2 \phi(X) \phi(X)^T \right]^{-1}$$

$$\cdot \widehat{\mathbb{E}}_{n_E} \left[ \left( YZ(1 - \pi_Z(X)) - Y(1 - Z)\pi_Z(X) - w(X)\tau^O(X) \right) w(X)\phi(X) \right]$$

$$\widetilde{\theta} = \mathbb{E} \left[ w(X)^2 \phi(X) \phi(X)^T \right]^{-1}$$

$$\cdot \mathbb{E} \left[ \left( YZ(1 - \pi_Z(X)) - Y(1 - Z)\pi_Z(X) - w(X)\tau^O(X) \right) w(X)\phi(X) \right]$$

We note that these quantities are well defined because $\mathbb{E} \left[ w(X)^2 \phi(X) \phi(X)^T \right]$ is invertible. This follows from the first and last conditions of Assumption 3, along with the stipulation in Assumption 1 that $\gamma(x) \neq 0$ for all $x$ in a set of non-zero measure. Using these definitions, we can write

$$\left\| \widehat{\theta} - \theta \right\|_2 = \left\| \widehat{\theta} - \widetilde{\theta}_{n_E} + \widetilde{\theta}_{n_E} - \widetilde{\theta} + \widetilde{\theta} - \theta \right\|_2$$

$$\leq \underbrace{\left\| \widehat{\theta} - \widetilde{\theta}_{n_E} \right\|_2}_{\lambda_1} + \underbrace{\left\| \widetilde{\theta}_{n_E} - \widetilde{\theta} \right\|_2}_{\lambda_2} + \underbrace{\left\| \widetilde{\theta} - \theta \right\|_2}_{\lambda_3} \qquad \text{(Triangle Inequality)}$$

We study these terms separately. We notice that $\lambda_2$ is just linear regression of the modified outcome $YZ(1 - \pi_Z(X)) - Y(1 - Z)\pi_Z(X) - w(X)\tau^O(X)$ on $\phi(X)$ using weights $w(X)$. Given the regularity conditions in Assumption 3 (which subsume the standard regularity conditions of linear regression), we have that $\lambda_2$ is $O_p(1/\sqrt{n_E})$. Then, consider the $\widetilde{\theta}$ term. We have:

$$\widetilde{\theta} = \mathbb{E} \left[ w(X)^2 \phi(X) \phi(X)^T \right]^{-1}$$

$$\cdot \mathbb{E} \left[ \left( YZ(1 - \pi_Z(X)) - Y(1 - Z)\pi_Z(X) - w(X)\tau^O(X) \right) w(X)\phi(X) \right]$$

$$= \mathbb{E} \left[ w(X)^2 \phi(X) \phi(X)^T \right]^{-1}$$

$$\cdot \mathbb{E} \left[ \left( YZ(1 - \pi_Z(X)) - Y(1 - Z)\pi_Z(X) - w(X)\tau(X) + w(X)\theta^T\phi(X) \right) w(X)\phi(X) \right]$$
$$\text{(Realizability of } b(X))$$

$$= \mathbb{E} \left[ w(X)^2 \phi(X) \phi(X)^T \mid \gamma(X) \neq 0 \right]^{-1} P(\gamma(X) \neq 0)^{-1}$$

$$\cdot \Big( \mathbb{E} \left[ (YZ(1 - \pi_Z(X)) - Y(1 - Z)\pi_Z(X) - w(X)\tau(X)) w(X)\phi(X) \mid \gamma(X) \neq 0 \right]$$

$$+ \mathbb{E} \left[ w(X)^2 \phi(X) \phi(X)^T \theta \mid \gamma(X) \neq 0 \right] \Big) P(\gamma(X) \neq 0)$$

$$= \mathbb{E} \left[ w(X)^2 \phi(X) \phi(X)^T \mid \gamma(X) \neq 0 \right]^{-1}$$

$$\cdot \mathbb{E} \left[ (YZ(1 - \pi_Z(X)) - Y(1 - Z)\pi_Z(X) - w(X)\tau(X)) w(X)\phi(X) \mid \gamma(X) \neq 0 \right] + \theta$$

$$= \mathbb{E} \left[ w(X)^2 \phi(X) \phi(X)^T \mid \gamma(X) \neq 0 \right]^{-1}$$

$$\cdot \mathbb{E} \left[ \left( \frac{YZ}{\pi_Z(X)\gamma(X)} - \frac{Y(1 - Z)}{(1 - \pi_Z(X))\gamma(X)} - \tau(X) \right) w(X)^2 \phi(X) \Big| \gamma(X) \neq 0 \right] + \theta$$
$$\text{(Since } \gamma(X) \neq 0 \text{ implies } w(X) \neq 0 \text{ by Assumption 3)}$$

$$= \theta. \qquad \text{(Lemma 1)}$$

Thus, $\tilde{\theta} = \theta$ which implies $\lambda_3 = 0$. We now tackle the $\lambda_1$ term. To streamline the exposition, let us introduce the following shorthand notation:

$$\widehat{Y}^{(k)} := YZ(1 - \widehat{\pi}_Z^{(k)}(X)) - Y(1 - Z)\widehat{\pi}_Z^{(k)}(X)$$

$$\widetilde{Y} := YZ(1 - \pi_Z(X)) - Y(1 - Z)\pi_Z(X)$$

$$\widehat{\Sigma}_K := \frac{1}{K}\sum_{k=1}^{K}\widehat{\mathbb{E}}_k\left[\widehat{w}^{(k)}(X)^2\phi(X)\phi(X)^T\right]$$

$$\Sigma_K := \mathbb{E}\left[\widehat{w}^{(k)}(X)^2\phi(X)\phi(X)^T\right]$$

$$\widehat{\Sigma} := \widehat{\mathbb{E}}_{n_E}\left[w(X)^2\phi(X)\phi(X)^T\right]$$

$$\Sigma := \mathbb{E}\left[w(X)^2\phi(X)\phi(X)^T\right]$$

We can then write the $\widehat{\theta} - \widetilde{\theta}_{n_E}$ as follows:

$$\widehat{\theta} - \widetilde{\theta}_{n_E}$$

$$= \widehat{\Sigma}_K^{-1}\frac{1}{K}\sum_{k=1}^{K}\widehat{\mathbb{E}}_k\left[\left(\widehat{Y}^{(k)} - \widehat{w}^{(k)}(X)\widehat{\tau}^O(X)\right)\widehat{w}^{(k)}(X)\phi(X)\right]$$

$$- \widehat{\Sigma}^{-1}\widehat{\mathbb{E}}_{n_E}\left[\left(\widetilde{Y} - w(X)\tau^O(X)\right)w(X)\phi(X)\right]$$

$$= (\widehat{\Sigma}_K^{-1} - \widehat{\Sigma}^{-1})\frac{1}{K}\sum_{k=1}^{K}\widehat{\mathbb{E}}_k\left[\left(\widehat{Y}^{(k)} - \widehat{w}^{(k)}(X)\widehat{\tau}^O(X)\right)\widehat{w}^{(k)}(X)\phi(X)\right] \qquad (\lambda_{1,1})$$

$$+ \widehat{\Sigma}^{-1}\frac{1}{K}\sum_{k=1}^{K}\Big(\mathbb{E}[(\widehat{Y}^{(k)} - \widehat{w}^{(k)}(X)\widehat{\tau}^O(X))\widehat{w}^{(k)}(X)\phi(X)]$$

$$- \mathbb{E}[(\widetilde{Y} - w(X)\tau^O(X))w(X)\phi(X)]\Big) \qquad (\lambda_{1,2})$$

$$+ \widehat{\Sigma}^{-1}\frac{1}{K}\sum_{k=1}^{K}(\widehat{\mathbb{E}}_k - \mathbb{E})\Big[\left(\widehat{Y}^{(k)} - \widehat{w}^{(k)}(X)\widehat{\tau}^O(X)\right)\widehat{w}^{(k)}(X)\phi(X)$$

$$- \left(\widetilde{Y} - w(X)\tau^O(X)\right)w(X)\phi(X)\Big] \qquad (\lambda_{1,3})$$

By Cauchy-Schwartz, we can bound the $\lambda_1$ term as

$$\lambda_1 = \left\|\widehat{\theta} - \widetilde{\theta}_{n_E}\right\|_2 \leq \sum_{i=1}^{3}\|\lambda_{1,i}\|_2,$$

where we used the $\lambda_{1,i}$ notation introduced in the preceding equation. We bound each of the $\lambda_{1,i}$'s separately. We let $\|A\|_F$ denote the Frobenius norm of the matrix $A$. Then, consider $\lambda_{1,1}$:

$$\|\lambda_{1,1}\|_2$$

$$\leq \left\|\widehat{\Sigma}_K^{-1} - \widehat{\Sigma}^{-1}\right\|_F \left\|\frac{1}{K}\sum_{k=1}^{K}\widehat{\mathbb{E}}_k\left[\left(\widehat{Y}^{(k)} - \widehat{w}^{(k)}(X)\widehat{\tau}^O(X)\right)\widehat{w}^{(k)}(X)\phi(X)\right]\right\|_2$$

$$\text{(Cauchy-Schwartz)}$$

$$= \left\|\widehat{\Sigma}_K^{-1}(\widehat{\Sigma} - \widehat{\Sigma}_K)\widehat{\Sigma}^{-1}\right\|_F \left\|\frac{1}{K}\sum_{k=1}^{K}\widehat{\mathbb{E}}_k\left[\left(\widehat{Y}^{(k)} - \widehat{w}^{(k)}(X)\widehat{\tau}^O(X)\right)\widehat{w}^{(k)}(X)\phi(X)\right]\right\|_2$$

$$\leq \left\|\widehat{\Sigma}_K^{-1}\right\|_F \left\|\widehat{\Sigma} - \widehat{\Sigma}_K\right\|_F \left\|\widehat{\Sigma}^{-1}\right\|_F \left\|\frac{1}{K}\sum_{k=1}^{K}\widehat{\mathbb{E}}_k\left[\left(\widehat{Y}^{(k)} - \widehat{w}^{(k)}(X)\widehat{\tau}^O(X)\right)\widehat{w}^{(k)}(X)\phi(X)\right]\right\|_2$$

$$= O_p\left(\left\|\widehat{\Sigma} - \widehat{\Sigma}_K\right\|_F\right) \qquad \text{(By the boundedness conditions in Assumption 3)}$$

Furthermore,

$$
\widehat{\Sigma} - \widehat{\Sigma}_K = \widehat{\mathbb{E}}_{n_E}\left[w(X)^2\phi(X)\phi(X)^T\right] - \frac{1}{K}\sum_{k=1}^K \widehat{\mathbb{E}}_k\left[\widehat{w}^{(k)}(X)^2\phi(X)\phi(X)^T\right]
$$

$$
= \frac{1}{K}\sum_{k=1}^K (\widehat{\mathbb{E}}_k - \mathbb{E})\left[\left(w(X)^2 - \widehat{w}^{(k)}(X)^2\right)\phi(X)\phi(X)^T\right]
$$

$$
+ \mathbb{E}\left[\left(w(X)^2 - \widehat{w}^{(k)}(X)^2\right)\phi(X)\phi(X)^T\right]
$$

$$
\Rightarrow \left\|\widehat{\Sigma} - \widehat{\Sigma}_K\right\|_F \le \frac{1}{K}\sum_{k=1}^K \left\|(\widehat{\mathbb{E}}_k - \mathbb{E})\left[\left(w(X)^2 - \widehat{w}^{(k)}(X)^2\right)\phi(X)\phi(X)^T\right]\right\|_F
$$

$$
+ \left\|\mathbb{E}\left[\left(w(X)^2 - \widehat{w}^{(k)}(X)^2\right)\phi(X)\phi(X)^T\right]\right\|_F
$$

$$
\le \frac{1}{K}\sum_{k=1}^K \sum_{i,j=1}^d \Big|\underbrace{(\widehat{\mathbb{E}}_k - \mathbb{E})\left[\left(w(X)^2 - \widehat{w}^{(k)}(X)^2\right)\phi(X)_i\phi(X)_j\right]}_{:=\delta_k}\Big|
$$

$$
+ \left\|w - \widehat{w}^k\right\|_{L_2} \mathbb{E}\left[\left(w(X) + \widehat{w}^{(k)}(X)\right)^2\left\|\phi(X)\phi(X)^T\right\|_F^2\right]^{1/2}
$$

(Holder's inequality)

By our boundedness assumptions, the second term yields an $O_p\left(\|w - \widehat{w}^k\|_{L_2}\right) = O_p(r_\gamma(n_E) + r_{\pi_Z}(n_E))$ term in the expression for $O_p\left(\|\widehat{\Sigma} - \widehat{\Sigma}_K\|_F\right)$. To analyze the first term, let $E_k$ represent the samples in the $k^{\text{th}}$ fold of the $E$ dataset. Then, $\delta_k \mid E_k$ has mean 0 since $\widehat{w}^{(k)}$ is independent from $E_k$ due to the $K$-fold sample splitting. Then, we can apply Chebyshev's inequality to obtain

$$
\delta_k \mid E_k = O_p\left(n_E^{-1/2}\, \mathbb{E}\left[\left(w(X)^2 - \widehat{w}^{(k)}(X)^2\right)^2 \phi(X)_i^2\phi(X)_j^2\Big|E_k\right]^{1/2}\right) = o_p(1/\sqrt{n_E})
$$

from the consistency assumptions for $\widehat{\gamma}^{(k)}, \widehat{\pi}_Z^{(k)}$ which translate into a consistency assumption for $\widehat{w}^{(k)}$. By the bounded convergence theorem, this implies that $\delta_k$ is also $o_p(1/\sqrt{n_E})$. Putting everything together, we obtain

$$
\|\lambda_{1,1}\|_2 = O_p(r_\gamma(n_E) + r_{\pi_Z}(n_E)) + o_p(1/\sqrt{n_E}).
$$

We now tackle $\lambda_{1,2}$:

$\lambda_{1,2}$

$$
= \widehat{\Sigma}^{-1}\frac{1}{K}\sum_{k=1}^K \left(\mathbb{E}[(\widehat{Y}^{(k)} - \widehat{w}^{(k)}(X)\widehat{\tau}^O(X))\widehat{w}^{(k)}(X)\phi(X)]\right.
$$

$$
\left. - \mathbb{E}[(\widetilde{Y} - w(X)\tau^O(X))w(X)\phi(X)]\right)
$$

$\|\lambda_{1,2}\|_2$

$$
\le \|\widehat{\Sigma}^{-1}\|_F \frac{1}{K}\sum_{k=1}^K
$$

$$
\cdot \sum_{i=1}^d \left|\mathbb{E}\left[\left(\widehat{w}^{(k)}(X)\widehat{Y}^{(k)} - w(X)\widetilde{Y} - \widehat{w}^{(k)}(X)^2\widehat{\tau}^O(X) + w(X)^2\tau^O(X)\right)\phi(X)_i\right]\right|
$$

$$
\le \|\widehat{\Sigma}^{-1}\|_F \frac{1}{K}
$$

$$
\cdot \sum_{k=1}^K\sum_{i=1}^d \left\|\mathbb{E}\left[\widehat{w}^{(k)}(X)\widehat{Y}^{(k)} - w(X)\widetilde{Y} - \widehat{w}^{(k)}(X)^2\widehat{\tau}^O(X) + w(X)^2\tau^O(X)\big|X\right]\right\|_{L_2}\|\phi(X)_i\|_{L_2}
$$

Since the $\|\phi(X)_i\|$'s are bounded by assumption and $\widehat{\Sigma}^{-1} \xrightarrow{P} \Sigma^{-1}$ from the continuous mapping theorem, it suffices to study the term $\mathbb{E}\left[\widehat{w}^{(k)}\widehat{Y}^{(k)} - w(X)\widetilde{Y} - \widehat{w}^{(k)}(X)^2\widehat{\tau}^O(X) + w(X)^2\tau^O(X)\big|X\right]$:

$$\left\|\mathbb{E}\left[\widehat{w}^{(k)}(X)\widehat{Y}^{(k)} - w(X)\widetilde{Y} - \widehat{w}^{(k)}(X)^2\widehat{\tau}^O(X) + w(X)^2\tau^O(X)\big|X\right]\right\|_{L_2}$$

$$\leq \|\mathbb{E}[Y \mid Z = 1, X]\pi_Z(X)\{\widehat{w}^{(k)}(X)(1 - \widehat{\pi}_Z^{(k)}(x)) - w(X)(1 - \pi_Z(X))\}\|_{L_2}$$

$$+ \|\mathbb{E}[Y \mid Z = 0, X](1 - \pi_Z(X))\{\widehat{w}^{(k)}(X)\widehat{\pi}_Z^{(k)}(x) - w(X)\pi_Z(X)\}\|_{L_2}$$

$$+ \|\widehat{w}^{(k)}(X)^2\widehat{\tau}^O(X) - w(X)^2\tau^O(X)\|_{L_2}$$

$$\lesssim \|\widehat{w}^{(k)} - w\|_{L_2} + \|\widehat{\gamma}^{(k)} - \gamma\|_{L_2} + \|\widehat{\pi}_Z^{(k)} - \pi_Z\|_{L_2} + \|\widehat{\tau}^O - \tau^O\|_{L_2}$$
$$\text{(Boundedness assumptions)}$$

$$\lesssim \|\widehat{\gamma}^{(k)} - \gamma\|_{L_2} + \|\widehat{\pi}_Z^{(k)} - \pi_Z\|_{L_2} + \|\widehat{\tau}^O - \tau^O\|_{L_2} \qquad \text{(Definition of } w(X))$$
$$\leq r_\gamma(n_E) + r_{\pi_Z}(n_E) + r_{\tau^O}(n_O)$$

where $\lesssim$ absorbs constants. Thus, $\|\lambda_{1,2}\|_2$ is $O_p\left(r_\gamma(n_E) + r_{\pi_Z}(n_E) + r_{\tau^O}(n_O)\right)$. Lastly, we note that $\lambda_{1,3}$ is the empirical process equivalent of $\lambda_{1,2}$ and thus, by leveraging sample splitting through arguments similar those used for the $\lambda_{1,1}$ term, we have that $\|\lambda_{1,3}\|_2$ is $o_p(1/\sqrt{n_E})$. Putting all $\lambda_{1,i}$ terms together, we have that $\lambda_1$ is $O_p\left(r_\gamma(n_E) + r_{\pi_Z}(n_E) + r_{\tau^O}(n_O)\right) + o_p(\sqrt{n_E})$. Recall that $\lambda_2$ is $O_p(1/\sqrt{n_E})$ and $\lambda_3 = 0$, we obtain the desired result:

$$\left\|\widehat{\theta} - \theta\right\|_2 = O_p\left(r_\gamma(n_E) + r_{\pi_Z}(n_E) + r_{\tau^O}(n_O) + 1/\sqrt{n_E}\right).$$

Given that $\|\widehat{\tau} - \tau\|_{L_2} = \|(\theta - \widehat{\theta})^T\phi(X) + (\tau^O(X) - \widehat{\tau}^O(X))\|_{L_2}$, we further have

$$\|\widehat{\tau} - \tau\|_{L_2} = O_p\left(r_\gamma(n_E) + r_{\pi_Z}(n_E) + r_{\tau^O}(n_O) + 1/\sqrt{n_E}\right)$$

by using the derived $\widehat{\theta}$ rates, the Cauchy-Schwartz inequality and the boundedness of $\|\phi(X)\|_2$ assumption. Our proof is now complete.

## B.4 Proof of Theorem 3

We first study the convergence rate of $\widehat{\tau}^O$ using the conditions of Theorem 3. Assume that $h^O$ and $\phi(x)$ solve the following joint optimization problem:

$$\widehat{h}^O, \widehat{\phi} = \arg\min_{h^O \in \mathbb{R}^d, \phi \in \Phi} \sum_{i=1}^{n_O} \left(\left(\frac{Y^O A^O}{\widehat{\pi}_A(X)} - \frac{Y^O(1 - A^O)}{1 - \widehat{\pi}_A(X)}\right) - (h^O)^T\phi(X^O)\right)^2$$

Then, $\widehat{\tau}^O(x) = (\widehat{h}^O)^T\widehat{\phi}(x)$. Thus, we write:

$$\left\|\tau^O - \widehat{\tau}^O\right\|_{L_2} \leq \left\|(h^O)^T\phi(X) - (\widehat{h}^O)^T\widehat{\phi}(X)\right\|_{L_2}$$

$$\leq \left\|(h^O)^T\phi(X) - (\widehat{h}^O)^T\phi(X)\right\|_{L_2} + \left\|(\widehat{h}^O)^T(\phi(X) - \widehat{\phi}(X))\right\|_{L_2}$$

$$\lesssim \left\|h^O - \widehat{h}^O\right\|_2 + r_\phi(n_O) \qquad \text{(Boundedness assumptions)}$$

We further expand the first term:

$$\left\|h^O - \widehat{h}^O\right\|_2 = \left\|\mathbb{E}[\phi(X)\phi(X)]^{-1}\mathbb{E}[\widetilde{Y}\phi(X)] - \widehat{\mathbb{E}}_{n_O}\left[\widehat{\phi}(X)\widehat{\phi}(X)\right]^{-1}\widehat{\mathbb{E}}_{n_O}\left[\widetilde{Y}\widehat{\phi}(X)\right]\right\|_2$$
$$\left(\widetilde{Y} := \frac{Y^O A^O}{\widehat{\pi}_A(X)} - \frac{Y^O(1 - A^O)}{1 - \widehat{\pi}_A(X)}\right)$$

$$\leq \left\|\mathbb{E}[\phi(X)\phi(X)]^{-1}\mathbb{E}[\widetilde{Y}\phi(X)] - \mathbb{E}\left[\widehat{\phi}(X)\widehat{\phi}(X)\right]^{-1}\mathbb{E}\left[\widetilde{Y}\widehat{\phi}(X)\right]\right\|_2$$

$$+ \left\|\mathbb{E}\left[\widehat{\phi}(X)\widehat{\phi}(X)\right]^{-1}\mathbb{E}\left[\widetilde{Y}\widehat{\phi}(X)\right] - \widehat{\mathbb{E}}_{n_O}\left[\widehat{\phi}(X)\widehat{\phi}(X)\right]^{-1}\widehat{\mathbb{E}}_{n_O}\left[\widetilde{Y}\widehat{\phi}(X)\right]\right\|_2$$

$$= O_p(r_\phi(n_O) + 1/\sqrt{n_O})$$

Table 1: Hyperparameters of models in simulated data experiments.

| Method | Model(s) | Algorithm | Hyperparameter | Value |
|---|---|---|---|---|
| Algorithm 1 | Compliance | Random Forest (`scikit-learn`) | max_depth | 3 |
| | | | min_samples_leaf | 50 |
| Algorithm 1 | Outcomes | Random Forest (`scikit-learn`) | max_depth | 5 |
| | | | min_samples_leaf | 5 |
| Algorithm 2 | Representation CATE Compliance | Neural Network (`PyTorch`) | activation | ELU |
| | | | hidden units | 2 |
| | | | network depth | 5 |
| | | | weight_decay | 0.02 |
| | | | optimizer | Adam |
| | | | learning rate | 0.01 |
| | | | batch size | 2000 |
| | | | epochs | 1000 |

Thus, $\left\|\tau^O - \widehat{\tau}^O\right\|_{L_2}$ is $O_p(r_\phi(n_O) + 1/\sqrt{n_O})$. Next, we build upon the insights provided by the Proof of Theorem 2. We note that we can apply the same analysis as in the Proof of Theorem 2 by using $\widehat{\phi}$ instead of $\phi$ and everything goes through except the $\lambda_3$ term which is not 0 since $\nu$ depends on $\phi$ and not $\widehat{\phi}$. Thus, the convergence rate of $\|\widehat{\nu} - \nu\|_2$ will be $O_p\left(r_\gamma(n_E) + r_{\pi_Z}(n_E) + r_{\tau^O}(n_O) + 1/\sqrt{n_E}\right) = O_p\left(r_\gamma(n_E) + r_{\pi_Z}(n_E) + r_\phi(n_O) + 1/\sqrt{n_E} + 1/\sqrt{n_O}\right)$ plus a term that depends on the deviation between $\widehat{\phi}$ and $\phi$. This term is given by:

$$\lambda_3 = \left\| \mathbb{E}\left[w(X)^2 \widehat{\phi}(X)\widehat{\phi}(X)^T\right]^{-1} \right.$$
$$\left. \cdot \mathbb{E}\left[\left(YZ(1 - \pi_Z(X)) - Y(1-Z)\pi_Z(X) - w(X)\tau^O(X)\right)w(X)\widehat{\phi}(X)\right] - \nu \right\|_2$$
$$= \left\| \mathbb{E}\left[w(X)^2\widehat{\phi}(X)\widehat{\phi}(X)^T \Big| \gamma(X) \neq 0\right]^{-1} \mathbb{E}\left[w(X)^2\widehat{\phi}(X)\phi(X)^T\nu \Big| \gamma(X) \neq 0\right] - \nu \right\|_2$$
$$\text{(Lemma 1)}$$
$$= \left\| \mathbb{E}\left[w(X)^2\widehat{\phi}(X)\widehat{\phi}(X)^T \Big| \gamma(X) \neq 0\right]^{-1} \mathbb{E}\left[w(X)^2\widehat{\phi}(X)(\phi(X) - \widehat{\phi}(X))^T\nu \Big| \gamma(X) \neq 0\right] \right\|_2$$
$$= O_p(r_\phi(n_O))$$

However, this term simply gets absorbed into $O_p\left(r_\gamma(n_E) + r_{\pi_Z}(n_E) + r_\phi(n_O) + 1/\sqrt{n_E} + 1/\sqrt{n_O}\right)$. Thus, we obtain the desired results:

$$\|\widehat{\nu} - \nu\|_2 = O_p\left(r_\gamma(n_E) + r_{\pi_Z}(n_E) + r_\phi(n_O) + 1/\sqrt{n_E} + 1/\sqrt{n_O}\right),$$

and

$$\|\widehat{\tau} - \tau\|_{L_2} = O_p\left(r_\gamma(n_E) + r_{\pi_Z}(n_E) + r_\phi(n_O) + 1/\sqrt{n_E} + 1/\sqrt{n_O}\right).$$

## C   Additional Experimental Details

### C.1   Simulation Studies

**Implementation Details:** The results for the parametric extension from Section 5.1 were generated on a consumer laptop equipped with a 13th Gen Intel Core i7 CPU. The execution took approximately 1.5 minutes using 20 concurrent workers. In contrast, the representation learning outcomes were derived using an NVIDIA Tesla T4 GPU on Google Colab [19]. The execution took roughly 1.5 hours, with half the time spent on Algorithm 2 and the other half on learning $\widehat{\tau}^E(x)$ over 100 iterations.

The Random Forest (RF) models used in Algorithm 1 employ the `RandomForestRegressor` and `RandomForestClassifier` algorithms from the `scikit-learn` [42] Python library. For the feed-forward neural networks within the representation learning component, we utilize the `nn` module from the `PyTorch` package [41]. Details regarding the hyperparameters for these models are provided in Table 1.

Table 2: MSE $\pm$ SD for estimators in high-dimensional DGP

|        | $\widehat{\tau}^O(x)$ | $\widehat{\tau}^E(x)$ | $\widehat{\tau}(x)$ |
|--------|----------------|----------------|----------------|
| $d = 5$  | $1.40 \pm 0.09$ | $3.97 \pm 1.21$ | $0.40 \pm 0.07$ |
| $d = 10$ | $3.25 \pm 0.15$ | $7.70 \pm 1.54$ | $1.25 \pm 0.20$ |
| $d = 20$ | $9.32 \pm 0.51$ | $19.2 \pm 2.58$ | $4.05 \pm 0.68$ |
| $d = 50$ | $37.1 \pm 0.94$ | $43.2 \pm 2.89$ | $9.39 \pm 1.64$ |

Table 3: 401(k) dataset description

| Name | Description | Type |
|------|-------------|------|
| age | age | continuous covariate |
| inc | income | continuous covariate |
| educ | years of completed education | continuous covariate |
| fsize | family size | continuous covariate |
| marr | marital status | binary covariate |
| two_earn | whether dual-earning household | binary covariate |
| db | defined benefit pension status | binary covariate |
| pira | IRA participation | binary covariate |
| hown | home ownership | binary covariate |
| e401 | 401(k) eligibility | binary instrument |
| p401 | 401(k) participation | binary treatment |
| net_tfa | net financial assets | continuous outcome |

We configured the parameters for the Random Forest (RF) models based on the theoretical guidance outlined in [44]. For the neural networks, we implemented early stopping using a validation dataset that constituted 20% of the total generated datasets.

**Result for High-Dimensional DGP:** We perform additional experiments to highlight the effectiveness of our method in higher-dimensional settings. To this aim, we modify the DGP in Section 5.1 to include $d$ features $X^d \in \mathbb{R}^d$, with both baselines and bias depending on all features as follows:

$$Y = 1 + A + X + 2A\beta^T X + 0.5X_1^2 + 0.75AX_1^2 + U + 0.5\epsilon_Y$$
$$U \mid X, A \sim N\left(\gamma^T X \left(A - 0.5\right), 0.75\right)$$

where the coefficients $\beta, \gamma \in [-1, 1]^d$ are set at random at the beginning of the simulation. In this setting, the bias function is given by $b(x) = -\gamma^T x$. We leave all other settings and parameters (including $n_O = n_E = 5,000$) unchanged and perform parametric extrapolation using Algorithm 1.

In Table 2, we report the mean squared error (MSE) and standard deviation (SD) of predictions on a fixed sample of 1,000 points drawn from the same distribution as $X$, over 100 iterations and for various dimensions ($d \in 5, 10, 20, 50$). The high MSE of the IV estimator $\widehat{\tau}^E(x)$ reflects the challenges of estimating compliance in high-dimensional settings. Likewise, the observational data estimator $\widehat{\tau}^O(x)$ shows clear bias. In contrast, the combined data estimator $\widehat{\tau}(x)$ from Algorithm 1 significantly outperforms both, demonstrating improved accuracy in this high-dimensional context.

### C.2 Impact of 401(k) Participation on Financial Wealth

**Implementation Details:** The dataset from [11] is comprised 9,915 observations with 9 covariates: age, income, education, family size, marital status, two-earner household status, defined benefit pension status, IRA participation, and home ownership indicators. We describe the features of the 401(k) dataset in Table 3.

Given the heavy-tailed distribution of net worth measures, we perform a pre-processing step to remove outliers. Specifically, we eliminate the top and bottom 2.5% of observations, effectively narrowing the range of potential outcomes from $[-0.5 \times 10^6, 1.5 \times 10^6]$ to $[-1.4 \times 10^4, 1.34 \times 10^5]$. This adjustment leaves us with 9,419 observations, which are then evenly distributed between the observational and experimental datasets. We find that this procedure improves the stability of regression and classification algorithms across different random data splits.

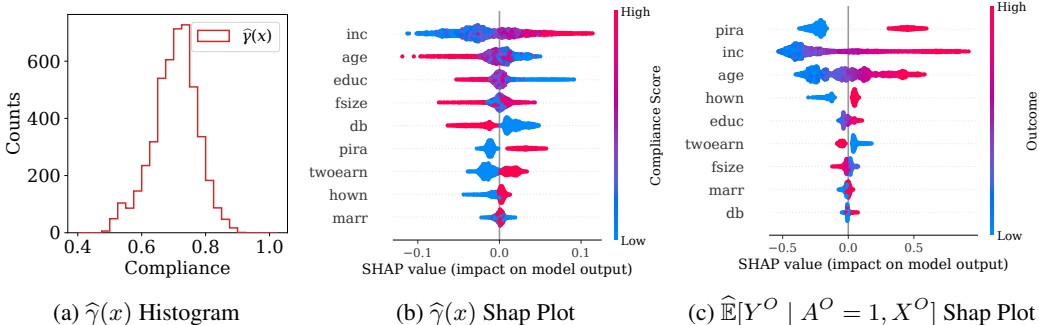

(a) $\widehat{\gamma}(x)$ Histogram  (b) $\widehat{\gamma}(x)$ Shap Plot  (c) $\widehat{\mathbb{E}}[Y^O \mid A^O = 1, X^O]$ Shap Plot

Figure 4: Characteristics of the 401(k) dataset derived from the first stage of Algorithm 1. (4a): Histogram of compliance scores for $x \in X^E$. (4b): Shapley plot [37] for the compliance model in the IV dataset with features arranged in decreasing order by feature importance. (4c): Shapley plot for the estimated outcome model $\widehat{\mathbb{E}}[Y^O \mid A^O = 1, X^O]$ in the observational dataset with features arranged in decreasing order by feature importance.

Table 4: MSE $\pm$ SD across different 401(k) data splits. Age: 40, Income: $30,000, Single

| Educ | $\widehat{\tau}^O$ (in 1,000$) | $\widehat{\tau}^E$ (in 1,000$) | $\widehat{\tau}$ (in 1,000$) |
|---|---|---|---|
| 8 | $11.9 \pm 2.18$ | $10.0 \pm 2.23$ | $9.83 \pm 2.22$ |
| 10 | $11.8 \pm 2.17$ | $10.2 \pm 2.42$ | $9.99 \pm 2.18$ |
| 12 | $11.8 \pm 2.22$ | $9.88 \pm 2.36$ | $10.2 \pm 2.20$ |

Table 5: MSE $\pm$ SD across different 401(k) data splits. Age: 40, Income: $30,000, Married

| Educ | $\widehat{\tau}^O$ (in 1,000$) | $\widehat{\tau}^E$ (in 1,000$) | $\widehat{\tau}$ (in 1,000$) |
|---|---|---|---|
| 8 | $11.3 \pm 2.40$ | $9.49 \pm 2.23$ | $9.54 \pm 2.50$ |
| 10 | $11.3 \pm 2.40$ | $9.63 \pm 2.40$ | $9.59 \pm 2.38$ |
| 12 | $11.2 \pm 2.41$ | $9.93 \pm 2.39$ | $9.65 \pm 2.29$ |

This dataset has previously been analyzed using Random Forest algorithms in [12]. Consistent with this earlier work, we employ the same models (`RandomForestRegressor` and `RandomForestClassifier` from `scikit-learn`) and use identical hyperparameters (n_estimators = 100, max_depth = 6, max_features = 3, min_samples_leaf = 10) for various regression and classification tasks outlined in Algorithm 1. For the second stage of Algorithm 1, we use a `Lasso` regressor from `scikit-learn` with a penalty of $\alpha = 0.07$ selected via 5-fold cross-validation.

In Figure 4, we display several characteristics of the 401(k) dataset derived from the first stages of Algorithm 1. In particular, we illustrate the spread in compliance scores in IV dataset, as well as the impact of important features on the predictions of the compliance and outcome models, respectively. As noted in the main text, the compliance scores are relatively large and range between 0.49 and 0.90 (mean=0.70). Furthermore, the primary features influencing the compliance score model include income, age, and education. In contrast, the features impacting the outcome model $\widehat{\mathbb{E}}[Y^O \mid A^O = 1, X^O]$ are IRA participation, income, and age, with education having a significantly lesser effect. This motivated us to investigate how education influences the derived CATEs.

**Quantifying Uncertainty Across Data Splits:** We quantify our claims for the 401(k) study by repeating the experiment across 100 different $(O, E)$ splits of the original data. We calculate the means and standard deviations of the treatment effects by years of education for the two examples described in the paper, with the results shown in Table 4 and Table 5. We note that the original trend (biased observational estimates, accurate extrapolation to the no-compliance region) is largely preserved, and our method demonstrates the ability to interpolate well on average in the artifically introduced non-compliance region. However, the uncertainty, as reflected by the large standard deviations, is substantial enough that the results are not statistically significant, which limits the strength of the conclusions we can draw from this experiment (unfortunately!). This is most likely

due to the prevalence of outliers, as net worth follows a heavy-tailed distribution, and RF regressors tend to overfit to these extreme values.

## D  Limitations and Societal Impacts of Our Work

Our methodology hinges on several key assumptions, and violations can significantly affect the accuracy and reliability of our estimates. First, the standard IV assumptions (Assumption 1) must hold. If the instrument directly affects the outcome, is correlated with unobserved confounders, or is weak across all strata of covariates, our estimates may be biased and unreliable. Some of these issues can be mitigated in experimental settings where the instrument is fully randomized. Additionally, the unconfounded compliance assumption requires that compliance is independent of potential outcomes given the covariates. Violations here can also lead to biased estimates if unrecorded explanatory variables affect both outcomes and compliance. Lastly, our method relies on realizability assumptions regarding the bias function. If these assumptions do not hold, our estimates might be biased.

The societal impacts of our method stem from potential inaccuracies in treatment effect estimates and their subsequent use. Inaccurate treatment effect estimates could lead to a range of adverse outcomes, from a diminished user experience on online platforms to less effective healthcare recommendations, economic and public policies. Furthermore, while accurate estimates can provide substantial benefits, they must be used responsibly to avoid unintended consequences such as privacy concerns or potential biases in decision-making. It is thus crucial to apply these methods with careful consideration of ethical implications and societal impacts.

