# OpenReview forum: "Estimating Heterogeneous Treatment Effects by Combining Weak Instruments and Observational Data"
_NeurIPS.cc/2024/Conference — NeurIPS 2024 poster_

### Official Review · Reviewer_UxMs · 2024-07-03

**Soundness:** 3
**Presentation:** 3
**Contribution:** 3
**Rating:** 6
**Confidence:** 3

**Summary:**

This paper introduces a robust two-stage framework leveraging observational and instrumental variable data to predict conditional average treatment effects (CATEs), addressing biases from unobserved confounders and low compliance.

**Strengths:**

- This paper studies leveraging observational data and encouragement data with low compliance to predict conditional average treatment effects (CATEs) accurately.

- This paper proposes a two-stage framework that first learns biased CATEs from observational data and subsequently applies a compliance-weighted correction using weak IVs.

- This paper demonstrates its utility on real data 401(k) participation on financial wealth.

**Weaknesses:**

**About the identification of CATE.**

- The relevance assumption in Assumption 1 cannot ensure $\gamma(x)>0$.

- The identification of CATE (Eq.(3) on line 121) would be violated when $P(A^E(1)|x) < P(A^E(0)|x)$ for some $x$. In such cases, even if Assumptions 1 and 2 hold, and $\mathbb{E}[A^E \mid Z^E=1, X^E=x] - \mathbb{E}[A^E \mid Z^E=0, X^E=x] < 0$,  $\gamma(x)$ would be zero and Eq.(3) with $\gamma(x)=0$ no longer holds.

- In lines 39-41, the paper provides examples: “certain users on digital platforms may disregard recommendations either altogether or of certain undesired content, and on mobile health platforms certain participants may ignore recommendations (e.g., taking 250 steps per hour) due to time constraints or disinterest.” However, in Eq.(3), the paper does not consider the possibility that $\mathbb{E}[A^E \mid Z^E=1, X^E=x] - \mathbb{E}[A^E \mid Z^E=0, X^E=x] = 0$. This raises concerns about the soundness of the paper regarding the assumptions and theorems.


- The paper seems to implicitly assume the monotonicity of instrumental variables. It is necessary for the authors to clearly state the assumptions required for the theory and provide a complete proof of identifiability (Eq.(3)).


**I will re-evaluate the soundness of this paper according to the responses of the authors.**

**About the related work.** In the presence of unmeasured confounding, there are numerous methods that use proxy variables to estimate heterogeneous treatment effects, including VAE-based methods [Causal effect inference with deep latent-variable models] and negative control methods [A selective review of negative control methods in epidemiology].

**Types:** Line 97 has an extra space. Line 103 should be revised to 'b(x), i.e.'.

**Questions:**

- Does the method proposed in this paper require the additive noise assumption?

- Do the observed variables X and latent U exhibit distribution shifts between Observational data and Experimental data? Are the distributions of X and U consistent across both datasets?

- Is it feasible to set all Z=1 in the experimental data? Given that observational data effectively corresponds to the part where Z=0.

---

> ### Author Rebuttal · Authors · 2024-08-07
>
> **Strengths**
>
> Thank you for your positive feedback. We appreciate your recognition of our work on leveraging observational and encouragement data with low compliance to predict CATEs, and for acknowledging the utility of our two-stage framework.
>
> **Re: CATE Identification**
>
> You are absolutely correct that in its current form, our work requires that monotonicity is added to Assumption 1. The issue stems from the fact that we noticed that monotonicity is not actually needed, but did not propagate the required changes to the rest of the paper. In order for the analysis to be fully sound without monotonicity, the following small and local changes need to be implemented in the current draft (and will be added to the final version):
>
> * Add defiance indicator to Assumption 2:
>
> *Assumption 2 (Unconfounded Compliance). Define the compliance and defiance indicators $C$ and $D$ as $C:=\mathbb{I}[A^E(1)>A^E(0)]$ and $D:=\mathbb{I}[A^E(1)<A^E(0)]$, respectively. Then, $Y^E(1)-Y^E(0)\perp C \mid X^E$ and $Y^E(1)-Y^E(0)\perp D\mid X^E$.*
>
> * Revise definition of $\gamma(x)$ right after Eq. (3)
> $$\gamma(x)=P(C=1 \mid X^E=x)-P(D=1\mid X^E=x)$$
>
> * Proof of Lemma 1 in Appendix B.1:
>       Replace $\gamma(x)> 0$ with $\gamma(x)\neq 0$.
>
> We will also add a derivation of Eq. 3 to the appendix. We believe that removing the monotonicity assumption strengthens our work without significantly altering the analysis.
>
> **Re: Related Work**
>
> There are several other works that tackle confounding in observational studies. The ones that target point estimation (rather than bounds) usually assume the existence of auxiliary data and/or structure (e.g. proxy variables, negative controls). We will extend our literature review discussion to include these works as well.
>
> **Re: Questions**
>
> * The paper does not require the additive noise assumption, although additive noise is one way in which we can ensure the validity of Assumption 1.
> * Yes, the distributions of X and U are the same across the two datasets as the datasets are assumed to be from the same population/environment. We will make this explicit.
> * Excellent observation! Under certain assumptions about the collection of the IV and observational data, the observational data can be treated as the $Z=0$ component of the IV study. This would allow us to set $Z=1$ in the experimental study, thereby potentially increasing the data available for analysis.

---

> > ### Comment · Reviewer_UxMs · 2024-08-11
> >
> > Thanks for your rebuttal. My concerns have been solved, I will update my score accordingly.

---

> > > ### Author Response · Authors · 2024-08-13
> > >
> > > Thank you for your feedback. I'm glad we could address your concerns, and we appreciate your reconsideration and score update.

---

### Official Review · Reviewer_Gwhh · 2024-07-04

**Soundness:** 3
**Presentation:** 3
**Contribution:** 3
**Rating:** 6
**Confidence:** 3

**Summary:**

The paper presents a novel method for estimating Conditional Average Treatment Effects (CATEs) by integrating weak instrumental variables (IV) and observational data. This method addresses the challenges of unobserved confounding and low compliance often encountered in causal inference studies. The proposed framework involves a two-stage process: initially, biased CATEs are estimated using observational data, and subsequently, a compliance-weighted correction is applied using IV data. This correction leverages the variability in IV strength across covariates to improve the accuracy of CATE estimation. The method's efficacy is validated through simulations and real-world applications, such as assessing the impact of 401(k) participation on financial wealth.

**Strengths:**

The paper introduces a two-stage framework designed to estimate CATEs by effectively combining observational data with weak instrumental variables.

The proposed method is adept at handling unobserved confounding in observational data and low compliance in IV data, including scenarios with zero compliance for some subgroups.

The effectiveness of the method is demonstrated through extensive simulations and real-world applications, such as evaluating the effect of 401(k) participation on financial wealth.

**Weaknesses:**

How to understand assumption 2? Is it correct about $Y^{E}(1) - Y^{E}(0) \perp\!\!\!\perp C \mid X^{E}$?

What is IV datasets?

How could there be IV in the experimental data? IV has a correlation with treatment, but according to Lemma 1, this instrumental propensity does not have any correlation with treatment.


In the experimental data, was the treatment not randomly assigned?

Experimental data are usually small samples or difficult to collect, which makes the proposed method difficult to apply.

**Questions:**

See weakness

**Limitations:**

Yes

---

> ### Author Rebuttal · Authors · 2024-08-07
>
> **Strengths**
>
> Thank you for your thoughtful review. We appreciate your positive feedback on our method for estimating CATEs by integrating weak instrumental variables and observational data, and your acknowledgment of its novelty and effectiveness.
>
> **Re: Questions**
>
> * Assumption 2 states that the treatment effect is independent of compliance given the observed features, meaning there are no unobserved confounders affecting both compliance status and potential outcomes. This standard assumption (see [1, 2]) is crucial for identifying the CATE, rather than the CLATE (conditional local average treatment effect), from data with instrumental variables. We will include further explanations about this in the final version.
> * An IV dataset is a dataset that includes an instrumental variable along with the treatment, features, and outcome. We refer to this as experimental data or an experimental dataset, as we assume the instrument is randomized (conditionally or marginally). Our focus is on an intent-to-treat/encouragement design (e.g., movie recommendation, exercise encouragement) that can be easily deployed.
> * From Assumption 1, the instrument is relevant in some covariate strata, meaning it affects treatment uptake. The instrument propensity, defined as $\pi_Z(x) = P(Z^E = 1 \mid X^E = x)$, is independent of the treatment and can be estimated directly from the data.
> * Since the experimental data refers to intent-to-treat data with an instrument, the actual observed treatment is not randomized and is influenced by each unit's compliance. In short, the instrument is randomized, but the treatment is not. As you correctly pointed out, experimental samples with fully randomized treatments are challenging to obtain due to ethical considerations, financial constraints, and the infeasibility of enforcing specific treatments. This challenge is precisely why we propose our method.
>
> Based on these suggestions, we will include more clarifying discussions in our final version.
>
> [1] Wang, L. and Tchetgen Tchetgen, E., 2018. Bounded, efficient and multiply robust estimation of average treatment effects using instrumental variables. Journal of the Royal Statistical Society Series B: Statistical Methodology, 80(3), pp.531-550.
>
> [2] Frauen, D. and Feuerriegel, S., 2022. Estimating individual treatment effects under unobserved confounding using binary instruments. arXiv preprint arXiv:2208.08544.

---

> > ### Comment · Reviewer_Gwhh · 2024-08-12
> >
> > Thank you for the rebuttal. My concerns have been addressed, and I wish to maintain my score.

---

> > > ### Author Response · Authors · 2024-08-13
> > >
> > > Thank you for your feedback. We’re glad we could address your concerns, and we appreciate you reviewing our rebuttal.

---

### Official Review · Reviewer_xHgN · 2024-07-07

**Soundness:** 3
**Presentation:** 3
**Contribution:** 3
**Rating:** 6
**Confidence:** 4

**Summary:**

The authors propose an approach for conditional average treatment effect estimation using instrumental variables, extending existing work in IVA to settings with weak instruments (i.e., low treatment compliance in some population subgroups). In particular, the approach leverages a two-stage estimation setup: first, a biased CATE estimate is computed from the observational data, then corrected via reweighting based on compliance to obtain a final estimate.

Overall, the paper provides good exposition and intuition for a novel approach, and backs it up with convincing theoretical analysis. The theoretical results are fairly insightful, but can be improved with a little more clarification (details below). The empirical results are not quite as well-motivated, and could also be improved with some clarification of the motivation and differentiation from past work. In particular, I felt that better justification of why these datasets/evaluations are the correct ones to test the proposed approach are needed for me to appreciate the paper’s contributions. Furthermore, I’m a little bit unsure if there are sufficient comparisons to baselines in inference under weak IV in the paper as-is (citations below).

**Strengths:**

1. The exposition is clearly written and provides good intuition on IVA (though I am writing from the perspective of someone that has working knowledge of IVA already).
2. The proposed approach is intuitive and well-motivated, with good theoretical properties. Weak instruments are an inevitable problem in instrumental variable methods, so proposals to reduce their downstream impact are a salient area of research.
3. The paper itself is quite clearly written and mostly easy to understand.

**Weaknesses:**

I'd love to see the following points addressed to clear up any misunderstandings on my part:
1. My biggest criticism is about the motivation and coverage of the empirical results. Coverage-wise, while I appreciate the comparison with a vanilla LATE estimator ($\tau^E$ if I followed correctly?), additional comparisons to baselines in learning with weak instruments would help strengthen the paper, such as the ones cited by the authors as closely related [1, 2] — it’s not clear to me the empirical lift provided by the proposed approach compared to these methods. For the rebuttal phase, in lieu of new results, perhaps a precise explanation of how the authors’ proposed approach shares similarities and differs with [1, 2] would be most helpful.
2. The analysis of the 401k dataset results is slightly imprecise — I’m not sure I buy that $\hat{\tau}(x)$ closely tracks with $\tau^E$; I don’t have a prior for what’s “close enough.” Could the authors clarify more precisely (1) the motivation behind experiments on the 401k dataset (beyond extension to real-world data), (2) and why the results show proof-of-concept for the proposed approach?
3. I have a couple of overarching concerns about the theoretical results. As a first-order comment, how do the generalization bounds of the proposed approach compare to similar bounds for IVA/what are the trade-offs compared to past bounds? As a second-order comment, if I know that $\tau^O$ will be biased (i.e., due to unobserved confounding), why is it imperative for estimation error w.r.t. $\tau^O$ to be low as well (maybe this is so that, given a good estimate of $\theta$ — we get a good estimate of $b(x)$, and therefore correct for the bias)?

Nits (points that would improve the paper in my opinion, but are not urgent):
1. There are a few critical derivations where the clarity could be improved somewhat. It took me quite some time to follow how Eq. 4 was derived — while it’s probably okay to reserve most details to the Appendix, some intuition about which terms are being substituted where would be helpful, and making it clear in Eq. 4 that you’re taking the squared difference (example-wise) of the pseudo-outcome of Eq. 3 (as fitted on the intention-to-treat dataset) and a reweighted version of $b(x) + \tau^O(x)$ as fitted on the observational dataset (since it is equal to $\tau(x)$ by definition).
2. I notice that $\hat{\tau}^E$ and $\hat{\tau}^O$ are fitted on different subsets of the data — somewhat reminiscent of cross-fitting based estimators (e.g., [3]). Out of curiosity, is it necessary to fit the two estimators on different data splits for the theoretical guarantees to hold?

[1] Abadie, A., Gu, J., & Shen, S. (2024). Instrumental variable estimation with first-stage heterogeneity. Journal of Econometrics, 240(2), 105425. \
[2] Coussens, S., & Spiess, J. (2021). Improving inference from simple instruments through compliance estimation. arXiv preprint arXiv:2108.03726. \
[3] Kennedy, E. H. (2023). Towards optimal doubly robust estimation of heterogeneous causal effects. Electronic Journal of Statistics, 17(2), 3008-3049.

**Questions:**

I think my largest concerns were expressed above in the Weaknesses. I’d be happy to raise my score with a thorough and precise response to especially 1-2. Addressing the following might help strengthen the paper even more, but I don’t consider them as high-priority.
1. I’m slightly confused about the interpretation of Lemma 1 — to me, it seems like we have simply chosen to use IPW to estimate the numerator in Eq. 3. Is this understanding correct? If so, why not consider alternative estimators for the numerator (e.g., doubly-robust methods such as [1])? This is not a huge issue — just making sure I parsed the equation correctly.
2. Did the authors consider/evaluate the sensitivity of the proposed approach to different instantiations of the base learners (e.g., something besides RF/T-learner)? This is not a dealbreaker but would be a nice result to have.
3. Re: “The weighting scheme in Equation 4 creates a weighted distribution…” (L172) — what is this a weighted distribution of? The paper goes on to claim that the difference between the weighted distribution and the target distribution creates a transfer learning problem, but I’m having a bit of trouble understanding what these distributions are. Are these distributions defined over the covariates?
4. Re: Assumption 3-4 (realizability of $b(x)$) — could the authors expand on why this assumption is a reasonable one to make, or point to some works that have made similar assumptions? I think I’ve seen versions of the other assumptions, so I buy those, but I’m not sure if assuming $b(x)$ is linear in the representation $\phi$ is too limiting. Similarly, in Sec. 4.2, there’s an assumption that $\tau^O$ and $\tau$ — one of which is biased — have some shared representation. Is there a more concrete reason for why this is reasonable?

[1] Kennedy, E. H. (2023). Towards optimal doubly robust estimation of heterogeneous causal effects. Electronic Journal of Statistics, 17(2), 3008-3049.

**Limitations:**

The authors address their limitations in the Appendix. As a tiny suggestion, I think it is important to note that the lack of unobserved confounding is statistically unverifiable yet necessary for many causal inference approaches. Overall, I find the discussion of the limitations to be complete and well-written. As a very, very minor nitpick, I do believe it important to acknowledge the limitations of an approach more up-front (often in the conclusion) — it does not detract from my appreciation of the method and relegating discussion of limitations to the Appendix does not feel quite right to me.

---

> ### Author Rebuttal · Authors · 2024-08-07
>
> **Strengths**
>
> Thank you for your encouraging feedback. We are pleased that you recognize the novelty of our approach for estimating conditional average treatment effects using IVs, particularly in scenarios with weak instruments. Your positive feedback on the clarity of our exposition, the intuition of our methods, and the robustness of our theoretical results is appreciated and encourages us to improve our work further.
>
> **Re: Comparison with Coussens & Spiess, J. (2021) and Abadie et al. (2024)**
>
> These works focus on improving the efficiency of the 2SLS estimator for the local average treatment effect (LATE) by incorporating weights based on heterogeneous compliance. Our work, however, aims to estimate the CATE $\tau(x)$, which coincides with the conditional LATE (CLATE) $\tau^E(x)$ under our assumptions. The compliance-weighting strategy in their works cannot be directly applied here and does not improve the efficiency of our CATE ratio estimator in Eq. (3). We build on the concept of compliance weighting to improve the estimator in Eq. (3) in scenarios of low or no compliance by weighting samples for bias correction using observational data, diverging significantly from these studies.
>
> **Re: 401(k) Dataset Analysis**
>
> We agree that the language around this analysis needs precision and will revise it in the final version. The motivation behind these experiments is to demonstrate that our method can interpolate accurately in regions of no compliance, which we artificially introduce. We chose the 401(k) dataset due to its high estimated compliance ($0.49-0.90$), allowing us to establish a reasonably accurate ground truth CATE using Eq. (3). The figures show projection onto one axis of heterogeneity (years of education), but the procedure involves learning the biased CATE using the full sample and bias-correcting across all covariates and their interactions (46 features total for $\phi(x)$). The statement that the learned CATE extension "tracks" the true CATE refers to the close alignment in this projection. To quantify our claims, we repeated the experiment over 100 data splits, calculating the means and standard deviations of the treatment effects by years of education for the two examples in the paper. These tables are included in the rebuttal PDF and will be in the final version.
>
> **Re: Theoretical Results**
>
> To our knowledge, this is the first work to estimate CATEs from IV data with weak or no compliance by leveraging an additional observational dataset. Thus, comparing our estimator's bounds with IV-only estimators is challenging/inappropriate. Our estimator reduces variance in Eq. (3) when compliance is low and provides valid estimates when compliance is zero, where Eq. (3) fails. This reduction in variance and handling of zero-compliance cases is evident in our simulations.
> You are correct that a good estimator for $\theta$ (and therefore $b(x)$) requires a reliable estimator for $\tau^O(x)$. We use the learned $\widehat{\tau}^O(x)$ with the IV dataset to determine $b(x)$, crucial for correcting the unobserved confounding bias in $O$. We will discuss these points in the paper.
>
> **Re: Nits**
>
> Eq. 4 follows from Lemma 1 by noting (as you also pointed out) $\tau(x)$ as $\tau^O(x)+b(x)$ from our assumptions. We will make this connection explicit. We emphasize that $E$ and $O$ are fundamentally different datasets. $E$ includes an instrument from an IV study or randomized experiment with non-compliance, while $O$ is an observational dataset that may have been collected passively. Depending on the collection method, $O$ could be considered part of an IV study with $Z=0$, but this is not necessarily the case.
>
> **Re: Questions**
>
> * Yes, we could consider other estimating equations that might reduce the variance of the estimator in Eq. (3) at the cost of estimating 4 additional nuisances (see [1] below). However, this approach will not address the variance caused by low compliance, our primary concern. It would only reduce variance from potentially poor compliance estimation.
> * Our method is agnostic to the types of estimators used for $\tau^O(x)$. We can include additional evaluations in the experimental section using other estimators, such as the doubly robust estimator. It is important to note that we do not expect these alternative estimators to reduce confounding bias, as it is asymptotically irreducible.
> * The distribution over covariates $X$ becomes the distribution over compliers’ covariates when weighted by compliance.
> * Yes, several works assume a joint representation between causal inference learning tasks to enhance generalization across tasks such as counterfactual learning and bias correction in observational samples (e.g., see [2, 3, 4, 5] below). We discuss [2] and [3] in the related works section of our paper. Additionally, [4] and [5] are foundational works in joint representation learning for causal inference. While it is possible to generalize beyond linear representations and consider $b(x) = h \circ \phi(x)$, where $h$ is a hypothesis class (see [3] for an example), this is beyond the scope of our current work.
>
> References:
>
> [1] Frauen, D. and Feuerriegel, S., 2022. Estimating individual treatment effects under unobserved confounding using binary instruments.
>
> [2] Kallus, N., Puli, A.M. and Shalit, U., 2018. Removing hidden confounding by experimental grounding. Advances in neural information processing systems, 31.
>
> [3] Hatt, T., Berrevoets, J., Curth, A., Feuerriegel, S. and van der Schaar, M., 2022. Combining observational and randomized data for estimating heterogeneous treatment effects.
>
> [4] Shalit, U., Johansson, F.D. and Sontag, D., 2017, July. Estimating individual treatment effect: generalization bounds and algorithms. In International conference on machine learning (pp. 3076-3085). PMLR.
>
> [5] Shi, C., Blei, D. and Veitch, V., 2019. Adapting neural networks for the estimation of treatment effects. Advances in neural information processing systems, 32.

---

> > ### Comment · Reviewer_xHgN · 2024-08-09
> >
> > Thanks for the response! I appreciate the detailed answers, and I think most of my misunderstandings have been cleared up. Here's where I stand now:
> >
> > **W1:** Ok, I think I partially follow this — just to double-check, am I correct in saying that we cannot apply such past works because they can assist in targeting $\tau(x)$ (which is equivalent to $\tau^E(x)$ in the setting considered), but one of the core contributions of the work is incorporating $\tau^O(x)$ (or rather, the observational dataset in general) into estimates of $\tau^E(x)$ (i.e., as stated at the top of Section 4)?
> >
> > **W2:** Thanks for the tables — yeah, I can definitely see that the error bars would overlap. This would be great to turn into a figure for a camera-ready/future revision.
> >
> > **W3:** The clarifications make sense and address my concerns. I can see why alternative bounds in the literature would be incomparable, since they operate in a completely different problem setting from the proposed work. On a 2nd glance, the derived bounds indeed depend on both parameters of the observational and experimental datasets, which makes a lot of sense.
> >
> > **Questions:** Thanks for the clarifications — they've cleared up the misunderstandings. Good connections to shared-representation based models as well; I think I was a tiny bit concerned about assuming so such shared structure such that the $\tau - \tau^O$ is *linear* in some $\phi(x)$; the cited shared-representation approaches (TarNET [4] and DragonNet [5]; using citation numbers from rebuttal) still "split" the shared representation into neural-net based heads for each counterfactual distribution, right? Overall, while weakening this assumption would be nice, it's not a dealbreaker.
> >
> > My assessment of the paper has definitely improved after re-evaluation — the detailed responses targeted my concerns and have helped me gain a better understanding + appreciation of this work. I'm upgrading my score to 6 (WA).

---

> > > ### Author Response · Authors · 2024-08-13
> > >
> > > Thank you for reconsidering and raising your score!
> > >
> > > **W1:** Yes, you are correct. Additionally, the other works target LATE, which is a compliance-weighted average of $\tau(x)$ over the population $\mathcal{X}$ (similar to ATE in observational studies). They use compliance weighting to reduce the variance of the LATE estimator under a homogeneous linear IV model, which we do not assume here.
> > >
> > > **W2:** We will include that figure in the camera-ready version.
> > >
> > > **Questions:** You’re right—while the shared representation is theoretically split for different heads, in most applications (e.g., [3]), the part after the split is typically just a linear transformation. We will discuss how to potentially weaken this assumption in the camera-ready version.
> > >
> > > Please let us know if there are any other questions we can address.

---

### Official Review · Reviewer_znEs · 2024-07-17

**Soundness:** 3
**Presentation:** 3
**Contribution:** 3
**Rating:** 6
**Confidence:** 4

**Summary:**

The paper tackles the problem of estimating CATE when unobserved confounding is present in an observation study but an IV experiments is accessible, though the instrument could be weak. The paper proposes a two stage framework to first learn a biased CATE from observational data and makes a bias correction using complican weighted IV samples. The paper then demonstrate the effectiveness using simulation studies and a real world example on 401k.

**Strengths:**

Originality: There is a lot of work on combining observational data with experimental data to better estimate causal estimands. This paper is the first to consider combining with an IV study with potentially weak instruments.

Quality: The paper has clear theoretical results covering two cases and both simulation study and real world data example.

Clarity: I found the paper easy to follow, with clear related work, contributions of the paper, motivation, theoretical results and experiments.

**Weaknesses:**

Experiments:
1. It would be better to have some baseline comparisons, for example some debiased CATE estimation methods;
2. The simulation study and the real world example both use an equal size IV/observational data. What would happen if we only have a much smaller sample size IV study? I believe this is more common in real life since observational data is cheaper to get.
3. It would be interesting to see what happens in high-dimensional settings.

**Questions:**

1. Do you have any results on violation of different assumptions? For example the realizibility.
2. Also see weakness on questions on experiments.

**Limitations:**

The authors list the limitations in the appendix.

---

> ### Author Rebuttal · Authors · 2024-08-07
>
> **Strengths**
>
> Thank you for your insightful feedback. We appreciate your recognition of our novel approach to combining an observational dataset with an IV study with weak instruments. We are glad that our efforts to ensure the paper is both rigorous and accessible are apparent.
>
> **Re: More Experiments**
>
> * Baseline comparisons: Debiased CATE estimation methods are not ideal comparisons as they address finite sample bias rather than confounding bias. The confounding bias, typically found in observational studies, is an asymptotically irreducible bias:
> $$
> b(x) = \mathbb{E}\left[(\mathbb{E}[Y^O\mid A^O=1, X^O, U] - \mathbb{E}[Y^O\mid A^O=0, X^O, U])\mid X^O=x\right] - (\mathbb{E}[Y^O\mid A^O=1, X^O=x]- \mathbb{E}[Y^O\mid A^O=0, X^O=x])
> $$
> where U denotes the unobserved confounders. To the best of our knowledge, the only types of methods that address confounding bias either (1) use latent variable models to recover unobserved confounders, often from noisy proxies or multiple/sequential treatments (see [1,2]), or (2) combine observational and randomized data with perfect compliance (see [3,4]). For (1), additional data (e.g. multiple treatments, nosy proxies) that enable the latent modeling might not be available, and even if available, the unconfoundedness condition given the additional data cannot be tested in practice. For (2), randomized data with perfect compliance is difficult to obtain in practice in the settings we discuss, either due to implementation bottlenecks, ethical considerations, financial constraints, ets, which is what our paper addresses by considering encouragement/IV designs. We will add this discussion to the literature review section.
>
> * Dataset sizes: The dataset sizes (denoted by $n_E$​ for the experimental/IV dataset and $n_O$​ for the observational dataset) do not need to be equal. In Theorems 2 and 3, we show how both sizes influence the algorithm's convergence rate. The theorems detail the dependency of the convergence rate on $n_E$​ and $n_O$, illustrating how a smaller IV dataset ($n_E$​) affects the convergence rate. Since this is not currently reflected in the experimental section, we have added additional experimental results in the rebuttal PDF. Specifically, in Figure 1, we performed parametric extrapolation simulations by varying the ratio $n_E/n_O$​ while keeping $n_O=10,000$ fixed throughout the experiments. We display the mean squared error $\pm$ standard deviation across 100 iterations. As expected, the error in the $\widehat{\tau}^E$ estimation increases significantly with smaller $n_E$​ (due to low compliance in the finite sample). However, the corrected $\widehat{\tau}$ still shows a smaller mean error than the observational $\widehat{\tau}^O$, and this error steadily decreases as the $n_E/n_O$ ratio increases. We will include these additional results in the experimental section of the appendix.
>
> * High-dimensional settings: We conducted additional experiments by modifying the data-generating process (DGP) to include $d=10$ features, with both baselines and bias depending on all features:
> $$
> Y = 1 + A + X + 2A\beta^T X + 0.5X^2 + 0.75AX^2 + U + 0.5\epsilon_Y $$
> $$
> U \mid X=x, A=a \sim N\left(\gamma^T x\left(a-\frac{1}{2}\right), 1-\left(a-\frac{1}{2}\right)^2\right)
> $$
> where the coefficients $\beta, \gamma\in [-1, 1]^{d}$ are set at random at the beginning of the experiment. In this scenario, the bias is given by $b(x)=-\gamma^T x$. We leave all other settings and parameters (including $n_O=n_E=5000$) unchanged and perform the parametric extrapolation described in the paper. For this high-dimensional setting, we obtain the following results for the mean squared error (MSE) and standard deviation (SD) across 100 iterations:
>
> | |$\widehat{\tau}^O(x)$ | $\widehat{\tau}^E(x)$| $\widehat{\tau}(x)$|
> |--|--|--|--|
> |MSE$\pm$SD|$3.25\pm 0.15$|$7.70\pm 1.54$|$1.25\pm 0.20$|
>
> The high MSE of the IV estimator ($\widehat{\tau}^E(x)$) indicates the difficulty in estimating compliance in high-dimensional settings. Similarly, the observational data estimator ($\widehat{\tau}^O(x)$) is clearly biased. However, the combined data estimator ($\widehat{\tau}(x)$) performs much better in this high-dimensional setting. We will include more results (such as varying $n_E$​ and $n_O$​) for the high-dimensional setting in the experimental section appendix.
>
>
> **Re: Other Questions**
>
> When realizability does not hold, our estimator may be inconsistent, exhibiting a bias that persists asymptotically. The magnitude of this bias depends on the extent to which realizability is violated, i.e., how far the true representation deviates from the assumed function class. In some cases, it may still be beneficial to perform this analysis despite uncertainty about realizability, as the bias from this violation could be substantially smaller than the confounding bias.
>
> References:
>
> [1] Kuzmanovic, M., Hatt, T. and Feuerriegel, S., 2021, November. Deconfounding Temporal Autoencoder: estimating treatment effects over time using noisy proxies. In Machine Learning for Health (pp. 143-155). PMLR.
>
> [2] Wang, Y. and Blei, D.M., 2019. The blessings of multiple causes. Journal of the American Statistical Association, 114(528), pp.1574-1596.
>
> [3] Kallus, N., Puli, A.M. and Shalit, U., 2018. Removing hidden confounding by experimental grounding. Advances in neural information processing systems, 31.
>
> [4] Hatt, T., Berrevoets, J., Curth, A., Feuerriegel, S. and van der Schaar, M., 2022. Combining observational and randomized data for estimating heterogeneous treatment effects. arXiv preprint arXiv:2202.12891.

---

> > ### Comment · Reviewer_znEs · 2024-08-11
> >
> > Thanks for your response. All my concerns have been resolved.

---

> > > ### Author Response · Authors · 2024-08-12
> > >
> > > Dear Reviewer znEs. Thank you for reading our rebuttal. As you write that it has answered all your questions, we would greatly appreciate if you would raise your score accordingly. Thank you. And do let us know if you have further questions -- we would do our best to answer them promptly. Thanks again for reviewing our submission.

---

### Author Rebuttal · Authors · 2024-08-07

We thank all our reviewers for their thoughtful comments and constructive feedback. We are encouraged by the consensus on the novelty and effectiveness of our method, as well as its theoretical and empirical contributions. We have addressed additional questions and concerns in individual responses to each reviewer. If there are any remaining or new questions, please let us know, and we will address them promptly.

---

### Decision · Program_Chairs · 2024-09-25

**Decision:**

Accept (poster)

**Comment:**

The paper develops a new method for instrumental variable (IV) regression in a setting with weak instruments. All reviewers agreed about the novelty and relevance of this work to the NeurIPS community. The authors managed to iron out a few initial misunderstandings and unclear points during the rebuttal. For example, we concur with the authors that the comparison with standard IV methods is not meaningful as the underlying setting is inherently different. Lastly, we strongly encourage them to include the promised action points (e.g., the clarifications) in the final version. One option is to, for example, include a table but I leave the eventual decision to the authors as to what is appropriate.